# Chromatin-informed inference of transcriptional programs in gynecologic and basal breast cancers

Hatice U. Osmanbeyoglu [1,2], Fumiko Shimizu [3,9], Angela Rynne-Vidal [4,9], Direna Alonso-Curbelo[5,9], Hsuan-An Chen [5], Hannah Y. Wen[6], Tsz-Lun Yeung[4], Petar Jelinic[7], Pedram Razavi[8], Scott W. Lowe[5], Samuel C. Mok[4], Gabriela Chiosis[3], Douglas A. Levine [7] & Christina S. Leslie [2]

Chromatin accessibility data can elucidate the developmental origin of cancer cells and reveal the enhancer landscape of key oncogenic transcriptional regulators. We develop a computational strategy called PSIONIC (patient-specific inference of networks informed by chromatin) to combine chromatin accessibility data with large tumor expression data and model the effect of enhancers on transcriptional programs in multiple cancers. We generate a new ATAC-seq data profiling chromatin accessibility in gynecologic and basal breast cancer cell lines and apply PSIONIC to 723 patient and 96 cell line RNA-seq profiles from ovarian, uterine, and basal breast cancers. Our computational framework enables us to share information across tumors to learn patient-specific TF activities, revealing regulatory differences between and within tumor types. PSIONIC-predicted activity for MTF1 in cell line models correlates with sensitivity to MTF1 inhibition, showing the potential of our approach for personalized therapy. Many identified TFs are significantly associated with survival outcome. To validate PSIONIC-derived prognostic TFs, we perform immunohistochemical analyses in 31 uterine serous tumors for ETV6 and 45 basal breast tumors for MITF and confirm that the corresponding protein expression patterns are also significantly associated with prognosis.

[1] Department of Biomedical Informatics, University of Pittsburgh School of Medicine, Pittsburgh, PA, USA. [2] Computational & Systems Biology Program, Memorial Sloan Kettering Cancer Center, New York, NY, USA. [3] Chemical Biology Program, Memorial Sloan Kettering Cancer Center, New York, NY, USA. [4] Department of Gynecologic Oncology and Reproductive Medicine—Research, Division of Surgery, The University of Texas MD Anderson Cancer Center, Houston, TX, USA. [5] Department of Cancer Biology and Genetics, Memorial Sloan Kettering Cancer Center, New York, NY, USA. [6] Department of Pathology, Memorial Sloan Kettering Cancer Center, New York, NY, USA. [7] Laura and Isaac Perlmutter Cancer Center, New York University Langone Medical Center, New York, NY, USA. [8] Department of Medicine, Memorial Sloan Kettering Cancer Center, New York, NY, USA. [9] These authors contributed equally: Fumiko Shimizu, Angela Rynne-Vidal, Direna Alonso-Curbelo. Correspondence and requests for materials should be addressed to H.U.O. (email: osmanbeyogluhu@pitt.edu) or to C.S.L. (email: cleslie@cbio.mskcc.org)

Cancers arise through the accumulation of genetic and epigenetic alterations that lead to widespread gene expression changes. Transcription factors (TFs) are instrumental in driving these gene expression programs, and the aberrant activity of TFs—induced downstream of activated oncogenic signaling or in concert with epigenetic modifiers—often underlies the altered developmental state of cancer cells and acquisition of cancer-related cellular phenotypes. Data-driven computational strategies may help to infer patient-specific transcriptional regulatory programs and to identify and therapeutically target the TFs that lead to cancer phenotypes. Ultimately, such strategies could be used to personalize therapy and improve patient outcomes.

While several successful methods have been proposed for learning patient-specific regulatory programs, most regulatory network inference approaches in cancer use expression data only[1] or at best rely on analysis of TF motifs in annotated promoter regions[2–4]. However, in a few cancers—notably luminal breast and prostate cancer—ChIP-seq analyses of key transcriptional regulators, estrogen receptor (ER), and androgen receptor (AR) respectively, in both cell line models[5,6] and tumors[7,8] have revealed the importance of enhancers distal to gene promoters in gene regulatory programs. Incorporating DNA sequence information at intronic and intergenic enhancers should therefore improve the modeling of transcriptional regulation in tumors. Leveraging epigenomic data from cell line models, while imperfect, provides a feasible means to make a potentially large advance in the computational dissection of dysregulated gene expression programs in tumors.

Extensive pan-cancer genomic analyses have shown that the same genes and pathways are targeted by somatic alterations across multiple tumor types. These results suggest that pan-cancer modeling of regulatory programs could also be informative, as similar TFs may be dysregulated across cancers. So far, however, methods for inferring patient-specific regulatory programs have been applied to one cancer type at a time[1,9]. Multitask learning (MTL) refers to machine-learning algorithms that learn models for different problems that share information and/or parameters and provides a statistical framework for learning patient-specific regulatory models across multiple cancers[10]. MTL can improve accuracy by making use of limited data (small sample sizes) in each task by sharing information through the common model. This is especially important when reconstructing regulatory networks from high-throughput data because the number of parameters to fit is very large relative to the number of samples. In addition, extensive training data from more common tumor types may be able to compensate for smaller sample sizes in similar but rarer cancers.

Large-scale cancer genomics projects such as The Cancer Genome Atlas (TCGA) and others have suggested molecular similarities between gynecologic cancers from different sites of pelvic origin and breast cancers[11]. Specifically, uterine serous carcinomas (UCS), high-grade serous ovarian carcinomas (HGSOCs), and triple negative breast cancers (TNBCs) share frequent *TP53* somatic mutations and widespread somatic copy number alterations[11]. HGSOCs and TNBCs also both display inactivation of similar DNA repair pathways. Though each gynecologic disease has a variety of histologic subtypes, the most common and aggressive tumors including HGSOCs (OV)[12], UCS[13], and the serous-like subset of uterine (UCEC)[14], as well as basal breast cancer[15] were studied by TCGA. These tumors all lack adequate treatment options for recurrent disease and accurate predictors of response and resistance. Inferring patient-specific transcriptional regulatory programs may identify and eventually enable therapeutic targeting of transcriptional mechanisms underlying gynecologic malignancies for individualized treatment.

To improve inferring regulatory programs across cancer types, we developed patient-specific inference of networks incorporating chromatin (PSIONIC), a MTL method that jointly models transcriptional networks for several related cancer types by leveraging chromatin accessibility data in representative cancer cell lines. More specifically, PSIONIC integrates regulatory sequence from ATAC-mapped promoters and enhancers from a panel of cancer cell lines with RNA-seq data from patient tumors in order to infer patient-specific TF regulatory activities. We apply our approach to 723 RNA-seq experiments from gynecologic and basal breast cancer tumors[12–15] as well as 96 cell lines[16], using a novel ATAC-seq data set for cell line models of these cancers. ATAC-seq data from cell lines allows us to incorporate DNA sequence information at intronic and intergenic enhancers to improve the modeling of transcriptional regulation from tumor data. Although much work has been done in regression-based inference of transcriptional regulation from cis-regulatory information in a single tumor type, we use MTL across different tumor types to jointly learn patient-specific regulatory models. Our analysis identifies key transcriptional regulators as well as new prognostic markers and therapeutic targets.

## Results

**Pan-cancer modeling of regulatory programs.** To systematically identify TFs that drive tumor-specific gene expression patterns across multiple cancer types, we developed the PSIONIC computational framework (Fig. 1a). We started with an atlas of chromatin accessible events derived from cell line models of the tumor types to be analyzed, using ATAC-seq profiling data ("Methods" section). We represented every gene by its feature vector of TF-binding scores, where motif information was summarized across all promoter, intronic, and intergenic chromatin accessible sites assigned to the gene (see the "Methods" section). Single-task learning (STL) of a patient-specific regulatory model simply learns the TF activities that predict normalized gene expression levels in each tumor independently, using regularized regression (Fig. 1b, see the "Methods" section). In PSIONIC, we instead adopted a MTL approach called GO-MTL[17] to represent patient-specific TF activity model vectors across multiple tumor types as linear combinations of latent regulatory programs, where both the coefficients in the linear combination and the latent models were learned jointly by regression against all the normalized tumor expression profiles (Fig. 1c, see the "Methods" section). The latent regulatory programs capture common TF-gene regulatory relationships across patients both within and between tumor types.

**Gynecologic and basal breast cancer ATAC-seq analysis.** To enable PSIONIC modeling for gynecologic and basal breast tumors, we first generated a reference chromatin accessibility atlas for uterine (endometrioid, serous, carcinosarcoma), ovarian serous, and basal breast cancers using a panel of 12 cancer cell lines representing these five tumor types using the assay for transposase-accessible chromatin with high-throughput sequencing (ATAC-seq). We assembled an atlas of ~282K reproducible accessibility regions for all cell lines, as well as tumor type-specific atlases ranging from ~93 to ~153K reproducible regions (Supplementary Table 1, see the "Methods" section). Principal component analysis (PCA) identified heterogeneity in the chromatin accessibility landscape in these gynecologic and basal breast cancer cell lines (Fig. 2a, Supplementary Data 1). Notably, ovarian and basal breast cancer cell lines displayed more similar chromatin accessibility profiles than most of the uterine cancer cell lines. Interestingly, for the two uterine carcinosarcoma cell lines, the copy number high SNU685 cell line clustered with ovarian

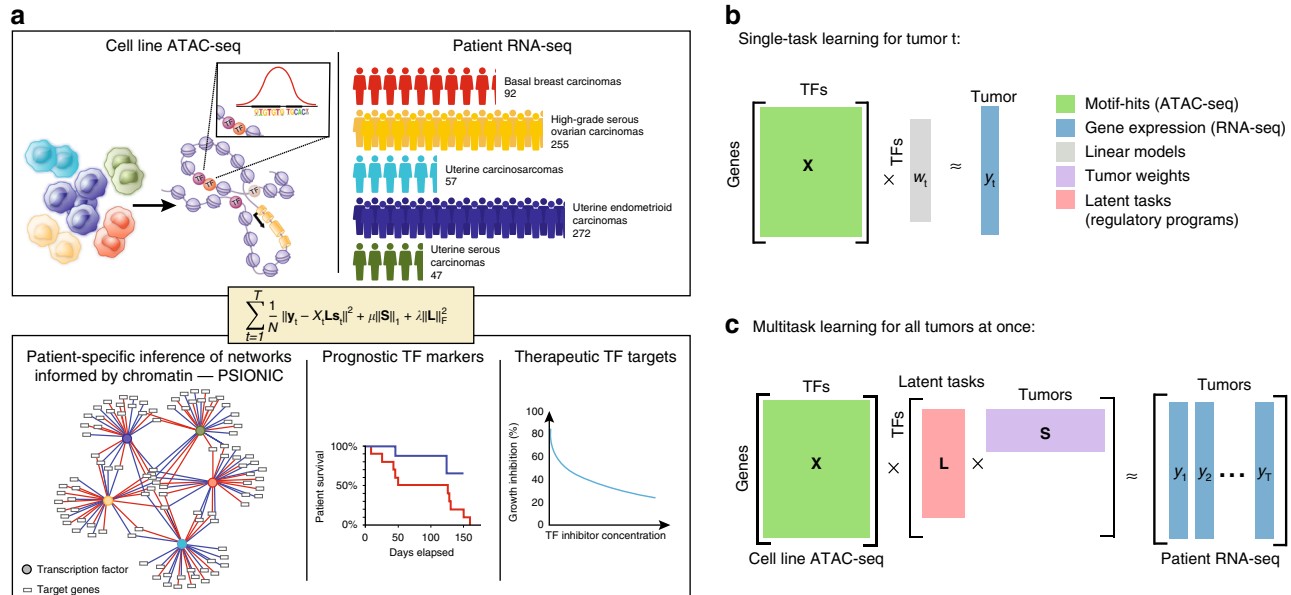

**Fig. 1** Overview of PSIONIC algorithm. **a** The input to our framework includes assay for transposase-accessible chromatin with high-throughput sequencing (ATAC-seq) profiles, TF recognition motifs, and tumor expression datasets. PSIONIC integrates regulatory information for each gene based on motifs in ATAC-mapped promoters and enhancers from cancer cell lines (**X**) with RNA-seq data from patient tumors (**Y**) in order to infer patient-specific TF activities (**W** = **LS**). Here, columns in the matrix **L** represent these latent regulatory programs, while **S**, the tumor weight matrix, captures the grouping structure and specifies the coefficients of the linear combination of latent regulatory programs for each tumor. A schematic comparison of **b** single task learning (STL) and **c** multitask learning (MTL) models

and basal breast cancer cell lines, whereas JHUCS1 clustered with uterine endometrioid cell lines.

Next, we assigned each accessible region in the tumor type specific atlas to the nearest gene (Fig. 2b), and we defined the regulatory locus complexity of a gene[18] as the total number of accessible regions within the tumor type. We grouped genes into three equally sized classes (tertiles) based on their regulatory complexity in tumor type specific atlases. Complexity classes were defined by dividing genes at the 33rd and 66th percentiles of the distribution of the number of accessible regions to produce groups with similar numbers of genes. We found that the normalized expression levels of low-complexity genes were lower than high-complexity and medium-complexity genes in tumor samples from TCGA for each tumor type ($P < 1 \times 10^{-16}$, one-sided Kolmogorov–Smirnov (KS) test for all comparisons). The importance of enhancers is illustrated by the region surrounding the *MDM2* gene. Despite the ubiquitous accessibility of the *MDM2* promoter, nearby distal regulatory elements of *MDM2* were more accessible in uterine endometrioid cell lines, consistent with higher *MDM2* gene expression observed in corresponding tumor samples from the TCGA cohort (Fig. 2c).

We also compared the cell line accessibility patterns with those of primary tumors using recently published ATAC-seq signal data for tumor samples from TCGA[19] including 13 UCEC-ENDO (24 with replicates) and 15 BRCA-BASAL (30 with replicates). Differential analysis of endometrial and basal breast cancer cell lines identified 366 endometrial-specific peaks and 368 basal breast-specific peaks (FDR $< 10^{-4}$, $\log_2$(FC) $> 3$). Consistent with our cell line data, high accessibility regions in breast cancer cell lines displayed significantly higher accessibility in BRCA-BASAL patients than in UCEC-ENDO ($P < 10^{-4}$, one-sided Wilcoxon signed-rank test, see the "Methods" section), while high accessibility regions in uterine endometrioid cell lines showed significantly higher accessibility in UCEC-ENDO patients than in BRCA-BASAL patients ($P = 0.00016$, one-sided Wilcoxon signed-rank test), as shown in Supplementary Fig. 1.

**Motifs underlying differential accessibility in cell lines**. Next, we determined the TFs that are most associated with open

chromatin for each tumor type through motif analyses and differential accessibility (see the "Methods" section). We examined the patterns of gain or loss of chromatin accessible regions between each pair of tumor types by performing pairwise differential read count analysis on accessible regions. The heatmap in Supplementary Fig. 2 shows the patterns of differential accessibility found among ~40,000 peaks across cell lines. Many TFs whose motifs were identified at differentially accessible regions between pairs of tumor types have roles in tumorigenesis (Fig. 2d, Supplementary Fig. 3, Supplementary Data 2). For example, chromatin peaks with HNF1 family motifs were more accessible in the endometrioid subset of uterine cell lines than in other cell types ($P < 10^{-16}$, one-sided KS test). HNF1β is associated with cancer risk in several tumors, including hepatocellular carcinoma, pancreatic carcinoma, renal cancer, ovarian cancer, endometrial cancer, and prostate cancer[20]. KLF and ETS family motifs were more accessible in endometrioid uterine and ovarian serous cell lines than in other cell types ($P < 10^{-16}$, one-sided KS test). These TFs have been implicated in the pathogenesis of these endocrine-responsive cancers of female reproductive tissues[21,22]. Chromatin peaks with FOS family motifs were more accessible in basal breast, ovarian serous and uterine carcinosarcoma and less accessible in uterine endometrioid cell lines than in other cell types ($P < 10^{-16}$, one-sided KS test). FOS family TFs have been implicated as regulators of cell proliferation, differentiation, and transformation.

In some cases the TF signal between cell lines might be due to the tissue of origin. To look more closely at this issue, we examined publicly available chromatin accessibility data in relevant normal tissues. We generated a reference chromatin accessibility atlas for normal uterine ($n = 1$), ovarian ($n = 3$), and breast ($n = 1$) tissue using DNase-seq data by the Roadmap Epigenomics project[23] and assembled an atlas of ~397K accessibility regions. We performed motif analysis in each chromatin accessible regions in the common atlas. Then, we examined the patterns of gain or loss of chromatin accessible regions between each pair of tumor types by performing pairwise differential read count analysis on accessible regions.

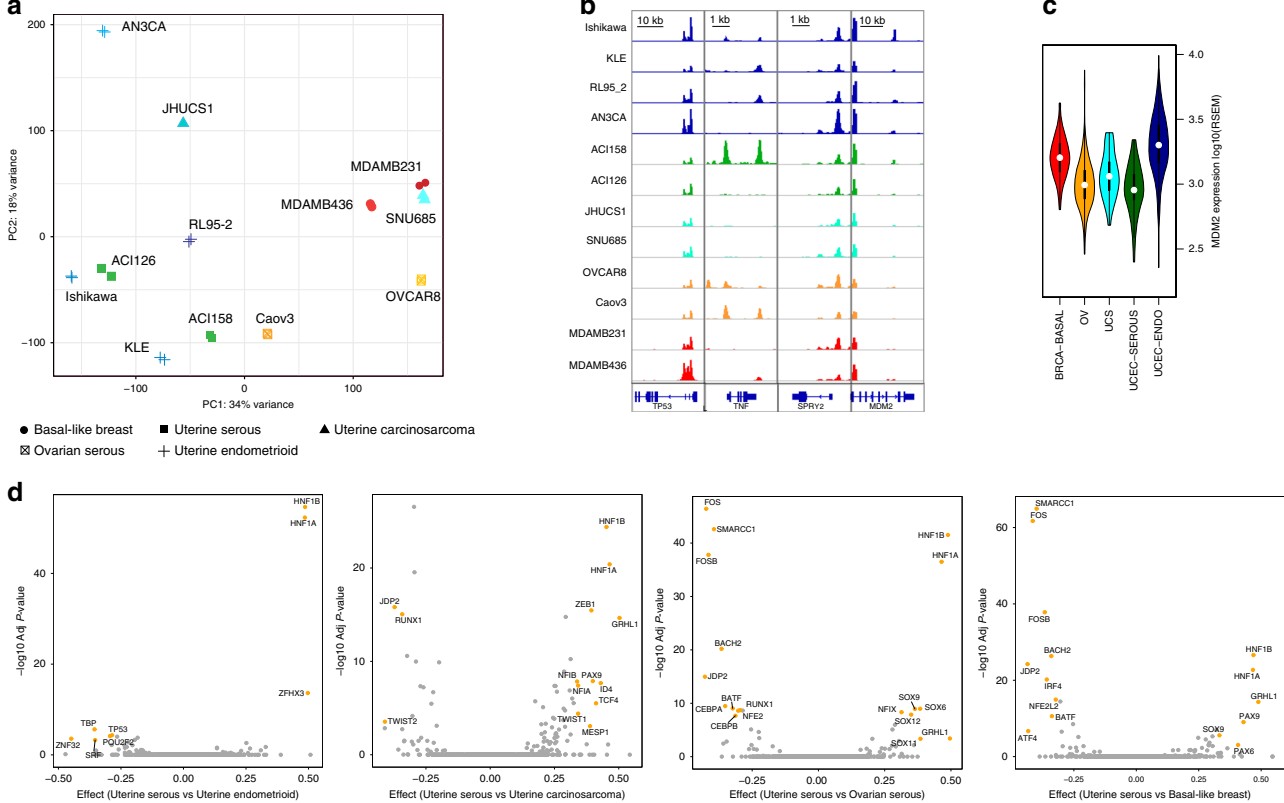

**Fig. 2** ATAC-seq analysis identifies key TFs in gynecologic and basal breast cancer cells. **a** Unsupervised principal component analysis based on the chromatin accessibility for all 12 cell lines at each of the 10K most variable chromatin accessible regions in the cell line panel. Samples are color coded according to the cell line type. Each symbol represents a single biological replicate and different symbols represent the tumor type of origin. Source data are provided as a Source Data file. **b** Normalized ATAC-seq profiles at important genes. Profiles represent the union of all biological replicates for each cell type. Genomic coordinates for the loci: *TP53*, chr17:7571720–7590868; *TNF*, chr6:31543344–31546112; *SPRY2*, chr13:80910112–80915086; *MDM2*, chr12:69201952–69239324. All y-axis scales range from 0 to 235 in normalized arbitrary units. The x-axis scale is indicated by the scale bars. **c** Violin plots indicate the distribution of *MDM2* gene expression across tumor types. *MDM2* gene expression is higher in uterine endometrioid carcinomas (UCEC-ENDO) compared to other tumor types. **d** Pairwise comparison of transcription factor motifs enriched in differentially accessible regions in cell lines. Volcano plot showing effect size versus –log10(adjusted P), using a Bonferroni correction to adjust P values for each plot. TF symbol annotations are written where the absolute value of the effect size is in at least top 30 and adjusted $P < 10^{-3}$. The foreground occurrence is the number of peaks containing a particular TF motif within the group of 5000 upregulated or 5000 downregulated peaks according to $\log_2$-fold-change read counts, respectively. The background occurrence is the number of peaks containing a particular TF motif found among all the differentially accessible peaks. Remaining pairwise comparisons are shown in Supplementary Fig. 3. Source data are provided as a Source Data file

While several FOS family motifs and SMARCC1 are enriched both in normal uterus vs. ovary as well as in the comparison of uterine serous vs. ovarian serous, in most cases the motifs identified by differential accessibility in cancer cell lines did not arise from the tissue of origin based on available normal tissue accessibility data (Supplementary Fig. 4). While many identified TFs are known to play a role in other cancers, their impact on gene regulation has not been characterized in gynecologic and basal breast cancers. We therefore developed a regression framework to model the regulatory role of TFs on gene expression in tumor samples.

**Multitask regression explains tumor expression profiles**. We next used a MTL strategy across tumor types to learn patient-specific regression models to predict tumor gene expression from gene regulatory sequence derived from cell line ATAC-seq data. Our method assumes that observed gene expression levels in each tumor can largely be explained by the unobserved activities of a smaller number of TF regulatory proteins through correlation with TF-binding motif scores. Moreover, our approach shares information across tumor samples and tumor types by representing each patient-specific regulatory model as a linear combination of a latent regulatory models.

Formally, we developed PSIONIC, a multitask-learning framework for integrating regulatory elements for each gene based on motifs in ATAC-mapped promoters and enhancers from cancer cell lines ($\mathbf{X}$) with RNA-seq data from patient tumors ($\mathbf{Y}$) to infer patient-specific TF regulatory activities ($\mathbf{W} = \mathbf{LS}$) (Fig. 1c). We adopted an algorithm for learning grouping and overlap structure in MTL (GO-MTL)[17]; here, the model does not assume a disjoint assignment of tasks (patients) to different groups (e.g. tumor type) but rather allows patient-specific models to overlap with each other by sharing one or more latent basis tasks, or latent regulatory programs. Further, the matrix $\mathbf{L}$ represents these latent regulatory programs, while $\mathbf{S}$, the tumor weight matrix, captures the grouping structure and specifies the coefficients of the linear combination of latent regulatory programs for each tumor. MTL enables selective sharing of information across other tumors, while standard STL trains a regression model for each tumor independently.

The application of our approach to 723 uterine, ovarian, and basal breast tumors from TCGA identified key TFs as potential common or cancer-specific drivers of expression changes. Our expression dataset included samples from five different tumor types, namely basal breast (BRCA-BASAL, $n = 92$), high-grade

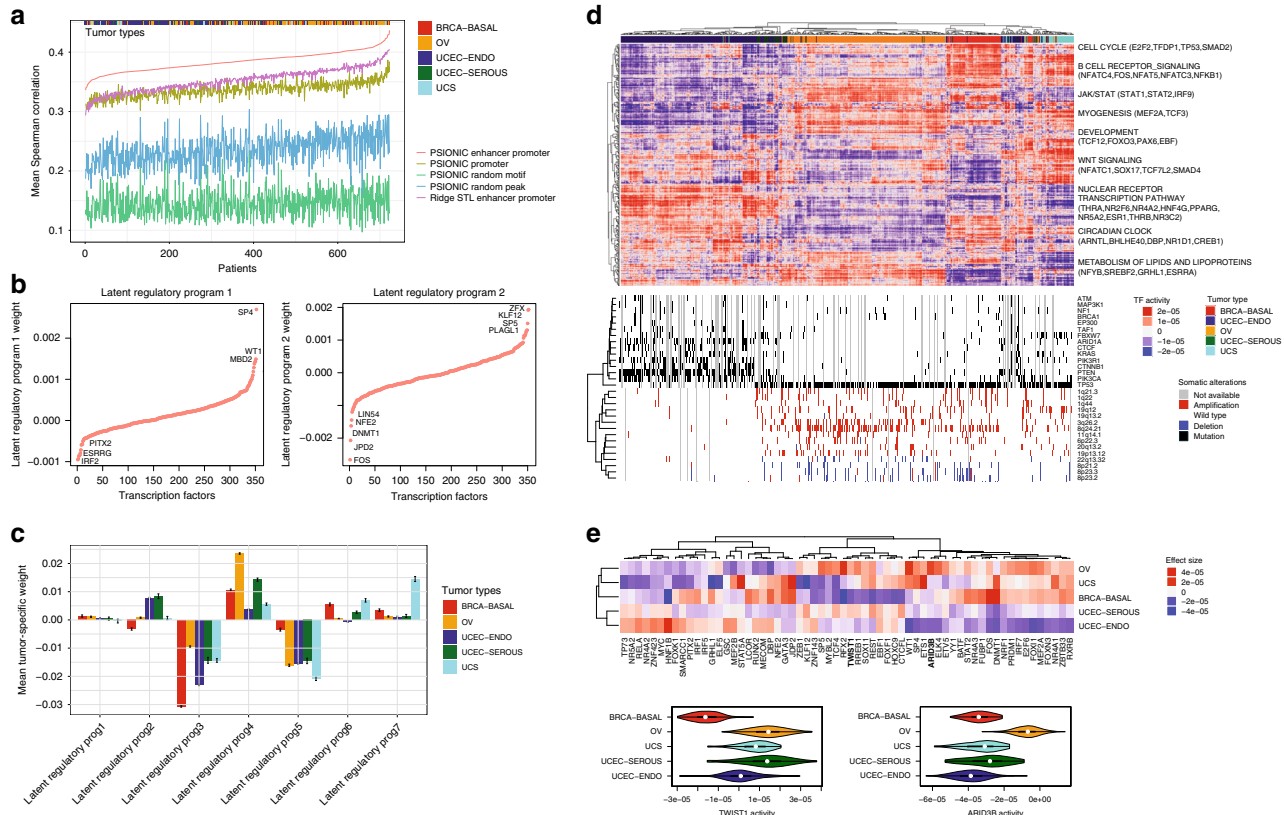

**Fig. 3** PSIONIC identifies regulatory features of tumor types. **a** PSIONIC and STL regression models predict differential expression of held-out genes and subtypes of tumor samples. Plot showing Spearman correlations between predicted and actual gene expression changes for all samples, sorted based on performance of the PSIONIC model using enhancer and promoter TF-binding sites. For each method and each sample, the Spearman correlation is computed using 10-fold cross-validation on held-out genes. Using TF-binding sites from enhancer promoter as features (mean $\rho = 0.384 \pm 0.016$) is significantly better than if we randomized motif hits for each chromatin accessible region across all motifs (mean $\rho = 0.144 \pm 0.022$; $P < 10^{-32}$, one-sided Wilcoxon signed-rank test), or if we randomized accessible regions for each motif, then assigned to the nearest gene (mean $\rho = 0.235 \pm 0.025$; $P < 10^{-32}$, one-sided Wilcoxon signed-rank test). PSIONIC models with motif data from promoter and enhancer regions outperformed models where only motif hits in promoter regions were used (mean $\rho = 0.337 \pm 0.012$; $P < 10^{-16}$, one-sided Wilcoxon signed-rank test) and STL approach based on ridge regression (mean $\rho = 0.352 \pm 0.019$; $P < 10^{-21}$, one-sided Wilcoxon signed-rank test). TCGA tumor types are shown in the top bar. **b** Example of latent regulatory programs. TFs are ranked based on the magnitude of coefficients. Remaining latent regulatory programs are shown in Supplementary Fig. 6. **c** Mean (± SE, standard error) tumor weight matrix (denoted by **S**) grouped according to tumor type for each latent regulatory program. **d** Hierarchical clustering mean-centered model vectors (denoted by **W**) on the TCGA tumor data sets. Source data are provided as a Source Data file. **e** Heatmap shows the mean inferred TF activity differences between samples in a given tumor type vs. those in all other tumor types. For each comparison, the absolute value of the mean inferred TF activity differences (effect sizes) are ranked and the union of top 20 TFs for each comparison are shown in the heatmap. Violin plots indicate the distribution of inferred ARID3B and TWIST1 TF activities across tumor types. Source data are provided as a Source Data file

serous ovarian (OV, $n = 255$), uterine carcinosarcoma (UCS, $n = 57$), uterine endometrioid carcinomas (UCEC-ENDO, $n = 272$), and uterine serous carcinomas (UCEC-SEROUS, $n = 47$). These results were obtained using binding site predictions for 352 human sequence-specific TFs based on motif hits from the Cis-BP database as motif data (see the "Methods" section).

Performance of PSIONIC and STL based on ridge regression for each tumor type using 10-fold cross-validation is shown in Fig. 3a. For statistical evaluation, we computed the mean Spearman correlation ($\rho$) between predicted and measured gene expression profiles on held-out genes for each tumor type and obtained mean $\rho = 0.384 \pm 0.016$ for PSIONIC, a highly significant result ($P < 10^{-16}$, one-sided Wilcoxon signed-rank test). This regression performance was significantly better than STL ($P < 10^{-21}$, one-sided Wilcoxon signed-rank test). Similarly, our models with motif data from promoter and enhancer regions outperformed models where only motif hits in promoter regions were used ($P < 10^{-16}$, one-sided Wilcoxon signed-rank test). By contrast, if we randomized motif hits for each chromatin

accessible region across all motifs, or if we randomized accessible regions for each motif, then assigned to the nearest gene, the prediction performance also significantly decreased ($P < 10^{-32}$, one-sided Wilcoxon signed-rank test).

When we compared 10-fold cross-validation results with different values of $K$, we found that prediction performance was stable after $K = 4$, with no sign of overfitting with higher $K$. However, a higher number of regulatory programs did allow PSIONIC-inferred models to better distinguish between tumors of distinct subtypes (Supplementary Fig. 5). Therefore, $K = 7$ seemed to be a reasonable choice for optimizing both overall prediction performance and capturing tumor-type-specific components of the regulatory models (Fig. 3b, Supplementary Fig. 6). Figure 3c shows a summary of mean tumor weights (**S**) across each tumor type for each latent regulatory program. For example, latent regulatory program 1 appeared to capture a common gene regulatory program shared across all cancer types, whereas latent regulatory program 2 captures TF-gene-regulatory relationships shared by uterine serous and endometrioid tumors.

Hierarchical clustering of tumors by inferred TF activities, $W = LS$, as derived from the model largely recovered the distinction between the major tumor types (Fig. 3d, Supplementary Data 3). In particular, clustering based on inferred TF activity mostly stratified patients by *TP53* mutation status. Uterine endometrioid tumors have distinct patterns of TF activities, consistent with their differing expression and mutational patterns.

**Multitask regression identifies tumor type-specific TFs.** Next, we assessed TF-tumor type associations by *t*-test to compare inferred TF activity between samples in a given tumor type vs. those in all other tumor types. We corrected for FDR across TFs for each such pairwise comparison and identified significant TF regulators and the results are shown in Supplementary Data 4 and Fig. 3e. FUBP1, which regulates *c-Myc* gene transcription, had significantly higher inferred activity in BASAL-BRCA than in gynecologic tumors, whereas ARID3B activity was significantly higher in OV, consistent with its role in promoting ovarian tumor development, in part by regulating stem cell genes[24]. NR5A2 (also known as liver receptor homolog-1, LRH-1) was significantly higher in uterine endometriod tumors, consistent with its function in regulating metabolism and hormone synthesis. Moreover, in agreement with previous reports, WT1 activity was significantly higher in ovarian serous[25] and uterine sarcoma[26]; TWIST1, a central player in the EMT, had increased activity in ovarian serous[27] and uterine serous cancers; YY1, which regulates various processes of development and differentiation and is involved in tumorigenesis of breast and ovarian cancer[28], had increased activity in these cancers.

In addition to confirming key TFs from previous studies, our analysis also predicted novel TF regulators in gynecologic and basal breast cancers. For example, MEF2A, a transcriptional regulator implicated in muscle development, cell growth control, and apoptosis, had significantly higher activity in OV, BRCA-BASAL, and UCS; the activity of microphthalmia-associated transcription factor (MITF), was significantly higher in uterine carcinosarcoma than in other cancers and displayed high variation across patients. Indeed, UCSs are characterized by an admixture of at least two histologically distinct components, one resembling carcinoma and another resembling sarcoma[13]. The roles of MEF2A and MITF have not been previously characterized in these cancers and may present promising targets for study and potentially for therapeutic intervention.

To investigate the potential of using PSIONIC-inferred TF activities to predict sensitivity to TF-targeted therapeutics, we decided to translate our model into cancer cell lines where drug sensitivity can be experimentally determined. Therefore, we first assembled a collection of basal-like, ovary and endometrium transcriptional profiles of immortalized human cancer cell lines from the CCLE[16], trained a PSIONIC model on this data set, and hence inferred cell line-specific TF regulatory activities. Similar to tumor models, we obtained significantly better regression performance with PSIONIC than with STL based on ridge regression in cell lines using 10-fold cross-validation (Supplementary Fig. 7). Regulatory models for cell lines to some extent recapitulated patient-specific tumor regulatory models (Supplementary Fig. 8). Importantly, cell line models as well as tumor models clustered mostly by cancer type.

While few drugs directly target TFs, we were able to use the metal-regulatory transcription factor-1 (MTF1) inhibitor LOR-253 for a proof of principle analysis. MTF1 is a ubiquitously expressed TF that is activated by heavy metals, redox stresses, growth factors, and cytokines[29]. We assessed our original panel of 10 cell lines for sensitivity to LOR-253 by measuring growth rate inhibition. Consistent with expectation, cell lines with higher inferred MTF1 activity showed a greater decrease in growth rate after the treatment with LOR-253 (Fig. 4). Overall, MTF1 inferred activity was significantly associated with growth rate inhibition by Spearman correlation analysis ($\rho = 0.795$ for these cell lines).

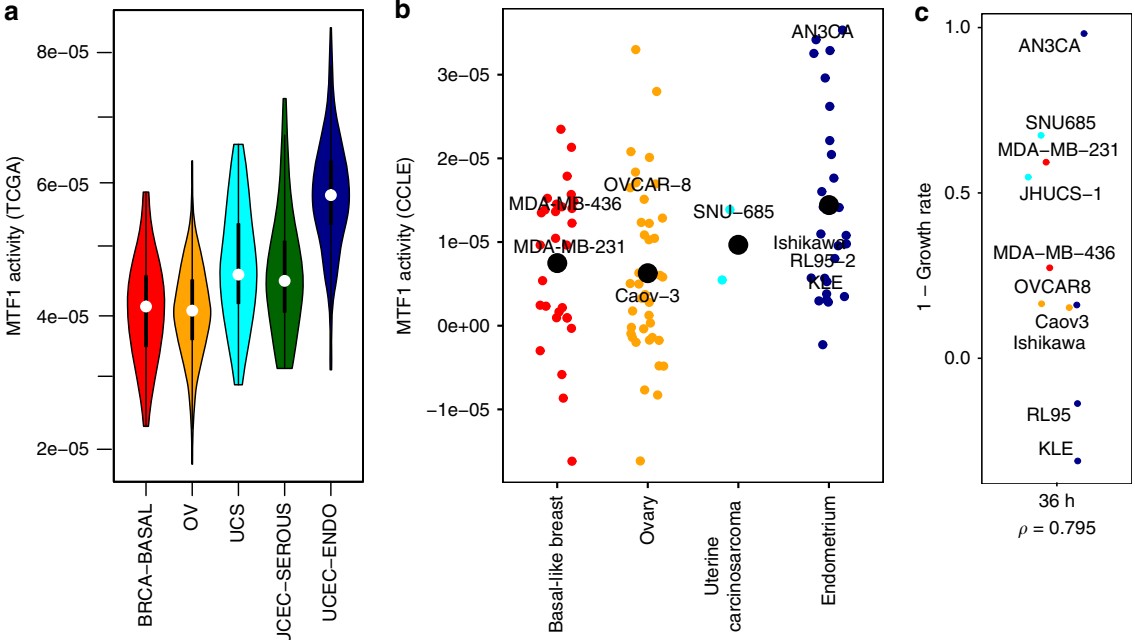

**Fig. 4** PSIONIC predicts cell line sensitivity to TF-targeted therapy. **a** Violin plots indicate the distribution of inferred MTF1 TF activities across tumor types. **b** We trained a PSIONIC model on 96 cell lines from the CCLE study. The dot plots show inferred MTF1 activities in basal-like breast, endometrium, ovary, and uterine carcinosarcoma cell lines. Black dots indicate mean inferred MTF1 activity for each tumor type. **c** 1—growth rate (GR) values[60] (growth inhibition) for MTF1 inhibitor LOR-253 36 h after the treatment in gynecologic and basal breast cancer cell lines (for these cell lines spearman correlation $\rho = 0.795$). Source data are provided as a Source Data file

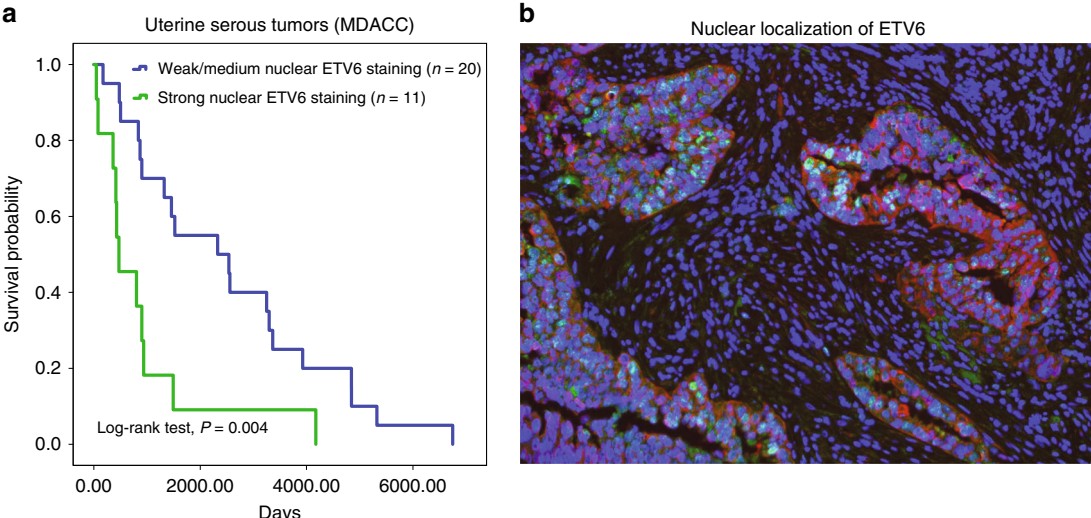

**Fig. 5** Clinical validation of ETV6 in uterine serous cancer. **a** Kaplan–Meier plot for uterine serous patients stratified by ETV6-staining score. Patient samples ($N = 31$) were divided into two groups based on intensity of ETV6 and positivity in nuclei or cytoplasm (patients with cytoplasmic, negative or weak/medium [score = 1 or 2] nuclear ETV6 staining, $N = 20$; and patients with strong [score = 3] nuclear ETV6 staining, $N = 11$). A significant difference in survival was observed between the groups ($P = 0.004$, log-rank test). The median survival was 2330 days (95% CI: 104–4556 days) for the cytoplasmic or weak nuclear group and 214 days (95% CI: 53–891) for the strong nuclear group. Source data are provided as a Source Data file. **b** Representative image of immunofluorescence staining on a primary uterine serous tumor. Double staining for ETV6 (green) and cytokeratin (red), shows nuclear localization of ETV6 in tumor cells; DAPI (4′,6-diamidino-2-phenylindole): blue. Source data are provided as a Source Data file

**Clinical outcome based on inferred TF activities**. To investigate the clinical relevance of TF activities, we examined whether inferred TF activities were associated with therapeutic response. The standard of care for ovarian serous patients is aggressive surgery followed by platinum/taxane chemotherapy. After therapy, platinum-resistant cancer recurs in ~25% of patients within 6 months[30]. The clinical significance of recurrence following current standard of care for ovarian serous patients prompted us to determine TFs linked to platinum resistance in OV. Inferred TF activities of seven TFs were significantly associated with platinum response, including HIF-1α and ZNF423 (t-test, $P < 0.05$, Supplementary Fig. 9). Consistent with our findings, HIF-1α has been associated with platinum resistance in a variety of cancers, including ovarian[31]. Moreover, ESR1 and ZNF423 have a role in cancer cell proliferation[32,33] and were significantly associated with platinum-sensitive tumors.

Next, we examined whether inferred TF activities were linked to survival data from the TCGA. We fit Cox proportional hazards regression models for each TF activity using clinical stage and age as additional covariates. The patient survival data and matched TF activities enabled us to perform TF-centric survival analyses to identify prognostic TFs within tumor type (TFs with FDR-adjusted $P < 0.02$, Cox analysis). Numerous TFs were significantly associated with survival outcome in BRCA-BASAL, UCEC-SEROUS, and UCEC-ENDO (Supplementary Tables 2–4). For some TFs, the prognostic value has been reported previously; for example, PGR[34] has been associated with survival in uterine cancer. However, most of the identified prognostic TFs lack prior reports of a link to survival in these cancers, making them potential candidates for follow-up studies.

For example, ETV6 inferred activity separated patients into high-risk and low-risk groups in UCEC-SEROUS (FDR < 0.02, Cox analysis). ETV6 exhibits antitumor effects suppressing proliferation and metastatic progression in prostate cancer[35]. However, its role in uterine serous cancer has not been studied. To further investigate whether prognostic TFs identified through inferred activity analyses could be verified at the protein level, we

performed immunohistochemical analyses in primary tumor samples from patients with uterine serous cancer ($n = 31$) for ETV6. Our analysis of the two groups of patient samples divided based on intensity of ETV6 and positivity in nuclei or cytoplasm showed a significant difference in survival between the groups ($P < 0.004$, log-rank test), with median survival of 2330 and 214 days for the weak or medium nuclear and the strong nuclear groups, respectively. The Kaplan–Meier survival curve based on ETV6 staining is shown in Fig. 5a. A representative image of immunofluorescence staining of a primary uterine serous tumor shows protein level nuclear localization of ETV6 in tumor cells (Fig. 5b).

Similarly, MITF inferred activity separated patients into high-risk and low-risk groups in BASAL-BRCA (FDR = 0.011, Cox analysis). Indeed, tissue microarray analyses in clinically annotated primary basal breast tumor samples ($n = 45$) validated MITF positivity in tumor cells and revealed a significant association between MITF expression and patient survival ($P < 0.006$, log-rank test), with median survival of 1208 and 2406 days for the positive and negative staining groups, respectively (see Kaplan–Meier survival curve and representative MITF-positive staining in basal breast cancer patients in Fig. 6a, b). MITF is a key TF in melanocyte development and differentiation and a diagnostic biomarker for metastatic melanoma[36]. However, the role of MITF in non-melanoma cancer cells, including basal breast cancer, is largely undefined. Thus, we next sought to functionally validate PSIONIC-predicted MITF activity in basal breast cancer cells.

To this end, we generated inducible shRNA vectors[37] targeting MITF and evaluated their impact on basal breast cancer gene expression. Potent shRNA-driven MITF downregulation was confirmed in both MDA-MB-436 basal breast cancer cells and SK-Mel-28 melanoma cells with known high MITF levels (Supplementary Fig. 10A–C). RNA-seq following MITF silencing revealed an effect on gene expression with 58 consistently downregulated and 103 consistently upregulated genes (adjusted $P < 0.05$ and fold change > 2) in MDA-MB-436 cells transduced with

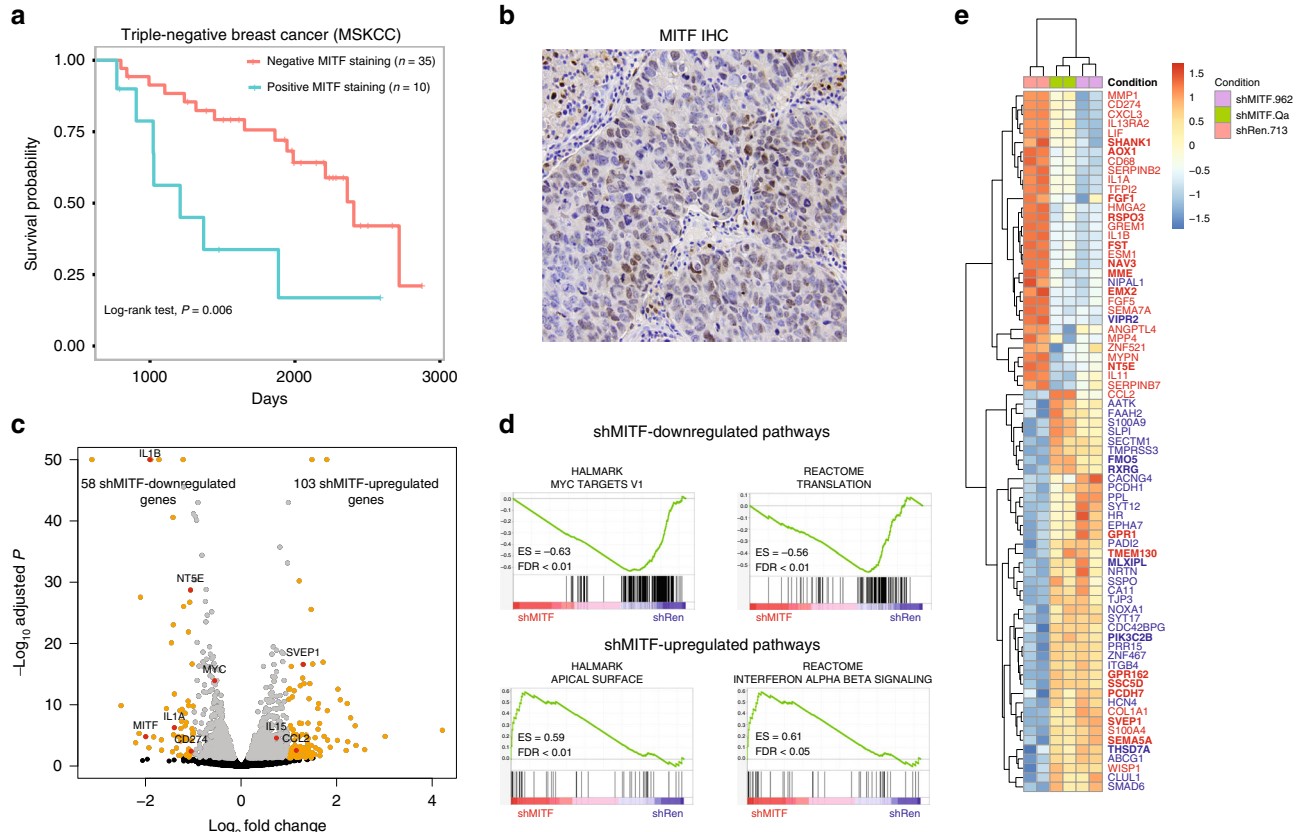

**Fig. 6** Clinical and in vitro validation of MITF in basal breast cancer. **a** Kaplan–Meier plot for basal breast cancer patients stratified by MITF staining score. Patient samples ($n = 45$) were divided into two groups based on MITF positivity ($n = 10$) and negativity ($n = 35$) in nuclei or cytoplasm staining. A significant difference in survival was observed between the groups (log-rank test, $P = 0.006$). The median survival was 1208 days for the positive staining group and 2406 days for the negative staining group. Source data are provided as a Source Data file. **b** Representative image of IHC staining with MITF antibody on a primary basal breast cancer tumor. **c** Volcano plot depicting the changes in representation ($\log_2$-fold change, x-axis) and significance ($-\log_{10}$ adjusted $P$, y-axis) of shRNA expressing MDA-MB-436 cells targeting Mitf vs. Ren at day 17. qPCR-validated genes (Supplementary Fig. 10) labeled. Source data are provided as a Source Data file. **d** Hallmarks of cancer and REACTOME gene sets analyzed from the transcriptome analysis comparing MDA-MB-436 cells transduced with two independent MITF shRNAs and control. Enrichment score (ES) is shown. **e** Heat map based on the subset of differentially expressed genes where gene expression correlated with PSIONIC-inferred MITF activity across breast cancer cell lines (target genes with $|\rho| > 0.4$ shown). Expression values were transformed (VST) and corrected. The color coding in each cell reflects the deviation from the gene's average across all samples. Red labels indicate positive correlation with inferred MITF activity, blue labels indicate negative correlation with inferred MITF activity. Bold labels indicate the existence of correlation in TCGA BASAL-BRCA tumors

two independent MITF shRNAs (Fig. 6c; Supplementary Table 5). Interestingly, commonly downregulated genes included c-Myc and c-Myc target genes, as well as additional pro-oncogenic factors, such as IL1B, NT5E (CD73), and other molecules with functions in tumor immune escape (Fig. 6d, Supplementary Tables 6 and 7)[38,39], which were validated by qPCR (Supplementary Fig. 10C). Commonly upregulated genes were enriched in ontology terms associated with immune activation (defensins, complement, IFN, IL15, CCL2) and cell adhesion (e.g. SVEP1) (Fig. 6d, Supplementary Tables 6 and 7, Supplementary Fig. 10D). These effects were not associated with changes in the proliferation rate of MDA-MB-436 cells in vitro yet are suggestive of an in vivo role for MITF in the regulation of cancer—microenvironment crosstalk in basal breast cancer. Importantly, most differentially expressed genes (DEGs) identified in MDA-MB-436 upon MITF suppression correlated with PSIONIC-inferred MITF activity across multiple basal breast cancer cell lines ($n = 29$; 75 out of 161 DEG, ~47%, $|\rho| > 0.4$, Fig. 6e) as well as across patient samples ($n = 92$; 43 out of 161 DEG, ~27%, $|\rho| > 0.4$). Together, these results validate the predictions made by PSIONIC on MITF activity and gene regulation in basal breast cancer.

## Discussion

With the development of high-throughput sequencing technologies, transcriptomic, proteomic, genomic profiles of tumor samples have been rapidly generated for diverse cancer types. Identifying differentially expressed genes or recurring mutations does not always clarify the molecular pathways that actually regulate tumor state and survival. There is still a large methodological gap between generating molecular profiles of tumor samples and understanding the molecular mechanisms underlying tumorigenesis and response to therapy.

Our PSIONIC method provides a systematic framework for integrating resources on regulatory genomics with tumor expression data to better understand gene regulation in cancers and infer patient-specific TF networks. PSIONIC uses a reduced rank representation model based on latent tasks, which helps regularize patient-specific regression models in light of noisy tumor gene expression data while sharing information between tumors and tumor types. Joint inference of TF activities across different tumor types may also reveal clinically relevant patient subgroups common to multiple cancers. As new ATAC-seq technologies for frozen tissue are developed[40], ATAC-seq will

become feasible in clinical samples, and then TF-binding site signals from tumor-specific ATAC-seq mapped regions can be incorporated to our framework.

One limitation of our approach is the multiplicity of inferred effects, which is biologically reasonable but complicates interpretation. Our model also currently makes the assumption that a TF either induces or represses its targets, but some TFs may play either role depending on coordination with co-factors. These limitations may confound the interpretation of inferred TFs with dual activator/repressor roles. Tumor data sets are also a challenging case for regulatory network analysis due to the presence of stromal/immune cells within the tumor and the heterogeneity of cancer cells themselves. However, the PSIONIC framework can be extended modeling of single-cell RNA-seq, as we will report elsewhere.

We used PSIONIC to perform a comprehensive transcriptional network analysis of gynecologic and basal breast cancer tumors. These tumors have not previously been subject to extensive epigenetic or computational analyses, and they all lack accurate predictors of response and treatment strategies for recurrent disease. PSIONIC can identify transcriptional processes that are active across otherwise very different tumors, such as MEF2A activity in the OV, BRCA-BASAL, and UCS cohorts. Applying our method to other pan-cancer cohorts such as squamous carcinomas or pediatric cancers might again find biological processes that are activated in a large number of tumor types and provide insight into common regulatory programs in tumors of different origin.

We demonstrated that PSIONIC-predicted activity for TFs in cell line models correlated with sensitivity to inhibition of a targetable TF, MTF1, giving a proof-of-principle for the potential therapeutic application of our approach. MTF1 target genes including PGF, HIF-1, and TGFB1 are involved in apoptosis, resistance, invasion, metastasis, and angiogenesis. Under normal conditions, MTF1 localizes both to the nucleus and the cytoplasm but accumulates in the nucleus upon these diverse stresses. After binding DNA, MTF1 recruits different co-regulators and often relies on other TFs, such as p300/CBP, Sp1, and HIF1α for coordinated target gene expression. Its established targets have important roles in metal homeostasis, embryonic development, tumor progression, and oxidative stress or hypoxia signaling. Importantly, inhibition of MTF1 induces the expression of tumor suppressor factor Kruppel like factor 4 (KLF4)[41]. This leads to the downregulation of cyclin D1, blocking cell cycle progression and proliferation. The MTF1 inhibitor LOR-253 enhances apoptosis induced by cisplatin in both SKOV3 and OVCAR3 cells[42], is cytotoxic to Raji and Raji/253R lymphoma cell lines[43], and suppresses the proliferation of acute myeloid leukemia (AML) cell lines[44]. A clinical trial testing LOR-253 in patients with AML and myelodysplastic syndrome is currently ongoing (ClinicalTrials.gov: NCT02267863). Our results suggest that the potential role of MTF1 in gynecologic and basal breast cancers merits further investigation.

We also showed that several PSIONIC-predicted TF activities were significantly associated with survival outcome in basal breast, uterine serous and endometrioid carcinomas. We validated two prognostic TFs, MITF and ETV6, in independent patient cohorts, giving a proof-of-principle for the potential prognostic application of our approach.

ETV6 encodes an ETS family transcription factor that is essential for hematopoietic processes[45]. Indeed, immunolocalization of ETV6 on tissue samples from uterine serous cancer patients demonstrated that strong nuclear ETV6 expression is significantly associated with poor disease prognosis. Possible future validation experiments to confirm the tumor-promoting roles of ETV6 in uterine serous cancer, expression levels of ETV6

can be manipulated in cultured cells through overexpression or silencing and the effects of ETV6 on cancer cell proliferation, survival, motility, and invasion potential can be evaluated. Ultimately, in vivo validation of the roles of ETV6 expression on uterine cancer progression can be studied using uterine serous cancer-bearing mouse models through in vivo silencing of ETV6 using siRNAs.

Encouraged by MITF's prognostic value in basal breast cancer patients, we directly examined the functional impact of loss of MITF in basal breast cancer cells by transducing MDA-MB-436 cells with inducible MITF shRNAs followed by RNA-seq. Although MITF shRNA did not compromise tumor cell proliferation in vitro, we found many cancer-relevant genes to be regulated by MITF, including cell-surface and secreted factors repressing immune-mediated anti-tumor responses in triple-negative breast cancer (e.g. NT5E/CD73)[38,39]. Moreover, many of the factors de-repressed upon MITF knockdown are important players that activate anti-tumor immunity (e.g. IL15, CCL2), which suggests a potential role of MITF in evasion of immune surveillance. Hence, our data suggest MITF might may play tumor-promoting roles in vivo by regulating the crosstalk of basal breast cancer cells with their tumor microenvironment. More globally, these analyses validate PSIONIC as a predictive tool to predict TF activity in specific tumor settings, as shown for MITF in basal breast cancer, that expands its role in cancer beyond its known lineage-specific functions in melanoma.

Patient-specific inference of TF networks may ultimately enable the development of individualized therapies, aid in understanding mechanisms of drug resistance, and allow the identification of biomarkers of response. We anticipate that computational modeling of transcriptional regulation across different tumor types will emerge as an important tool in precision oncology, aiding in the eventual goal of choosing the best therapeutic option for each individual patient.

## Methods

**Datasets**. RNA-seq data for each of the five tumor types were downloaded from TCGA's Firehose data run [https://confluence.broadinstitute.org/display/GDAC/Dashboard-Stddata]. Log10-transformed RNA-seq RSEM gene expression values were unit-normalized by tumor sample. Cancer cell lines RNA-seq data were downloaded from the CCLE website [http://www.broadinstitute.org/ccle]. $Log_{10}$-transformed RNA-seq TPM gene expression values were unit-normalized by cell line.

Bigwig files of ATAC-seq profiles of tumor samples from TCGA[19] including 13 UCEC-ENDO (24 with replicates) and 15 BRCA-BASAL (30 with replicates) were downloaded from https://gdc.cancer.gov/about-data/publications/ATACseq-AWG.

**Cell line selection for ATAC-seq**. In this study, we chose cell lines widely used as representative of corresponding tumor types depending on availability to our group. In several cases, we generated the first epigenomic characterization of these cell line models. ATAC-seq libraries generated from basal breast (MDA-MB-231, MDA-MB-436) high-grade serous ovarian (OVCAR8, Caov3), uterine carcinosarcoma (JHUCS, SNU685), endometrial endometrioid (AN3-CA, KLE, Ishikawa, RL95-2), and serous carcinoma (ACI-126, ACI-158) cell lines. Gynecologic cell lines OVCAR8, Caov3, JHUCS, SNU685, AN3-CA, KLE, Ishikawa, and RL95-2 were supplied by Douglas A. Levine. Uterine serous cell lines ACI-126 and ACI-158 were kindly supplied by John I. Risinger from Michigan State University. Basal breast cancer cell lines MDA-MB-231 and MDA-MB-436 were acquired from ATCC. The cell lines have been tested negative for mycoplasma contamination.

Briefly, Ishikawa and RL-95-2 derived from type I and KLE and AN3CA derived from type II endometrial carcinomas tumors have been widely used as models to investigate molecular genetics and mechanisms underlying their development, progression, and response to therapeutics[46]. KLE and AN3CA cells, originating from peritoneal and lymph node metastases, respectively, and RL-95-2 cells derived from a moderately differentiated (Grade 2) endometrial adenosquamous carcinoma. Ishikawa cells were established from the epithelial component of a moderately differentiated, stage 2, endometrial adenocarcinoma. CAOV3 and OVCAR8 have been widely used as representatives of high-grade serous cancer. CAOV3 and OVCAR8 possess TP53 mutations and substantial copy-number changes, key characteristics of high grade serous ovarian cancer (HGSOC). ACI-158 and ACI-126 are the main uterine serous (UPSC) cell lines. JHUCS-1 was established from a carcinosarcoma (malignant mixed

mesodermal tumor) of the uterus that was surgically removed from a 57-year-old Japanese woman[47]. SNU-685 was derived from uterine malignant mixed mullerian tumor[48].

**Sample preparation for ATAC-seq**. Cell lines were re-suspended in cold PBS according to ATAC-Seq protocol[49]. Chromatin was extracted and processed for Tn5-mediated tagmentation and adapter incorporation, according to the manufacturer's protocol (Nextera DNA sample preparation kit, Illumina®) at 37 °C for 30 min. Reduced-cycle amplification was carried out in the presence of compatible indexed sequencing adapters. The quality of the libraries was assessed by a DNA-based fluorometric assay (Thermo Fisher Scientific™) and automated capillary electrophoresis (Agilent Technologies, Inc.). Sample preparation and sequencing for ATAC-seq was performed by Epinomics. For each sample, ATAC-seq was performed on two biological replicates.

**ATAC data analysis**. Starting from fastq files containing ATAC-seq paired-end reads, sequencing adaptors were removed using Trimmomatic[50]. Trimmed reads were mapped to the hg19 human genome using Bowtie2[51] allowing at most 1 seed mismatch and keeping only uniquely aligned reads. Duplicates were removed using Picard (http://picard.sourceforge.net). For peak calling the read start sites were adjusted (reads aligning to the $+/-$ strand were offset by $+4$ bp/$-5$ bp, respectively) to represent the center of the transposase binding-event[49].

BigWig files were generated using bamCoverage from the deepTools suite with options—binSize 10–normalizeTo1 × 2451960000 –ignoreForNormalization chrX. The log$_2$-transformed ATAC-seq signal were calculated using bamCompare from deepTools[52]. Resulting normalized BigWig files were used as input to computeMatrix to calculate scores for regions of interest (using either scale-regions or reference-point mode) and visualized using plotHeatmap tool from deepTools.

Peak calling was performed on each cell type individually: first, the reads from different replicates were pooled, and the MACS2.0 peak caller[53] was then used to identify peaks with a permissive threshold ($P < 2 \times 10^{-3}$). Finally, IDR was used to identify reproducible peaks using two biological replicates for each cell type (IDR $< 1 \times 10^{-2}$). Peaks found reproducibly in each cancer cell subtype were combined to create a genome-wide atlas of accessible chromatin regions. Reproducible peaks from different samples were merged if they overlapped by more than 75%. The atlas of chromatin accessibility across 12 gynecologic and basal breast cancer cell lines contains 282,248 peaks. The number of reproducible peaks for each cell line and number of peaks in each cancer type specific atlas are listed in Supplementary Table 1.

We associated each peak to its nearest gene in the human genome using the ChIPpeakAnno package[54]. ATAC-seq peaks located in the body of the transcription unit, together with the 100 kb regions upstream of the TSS and downstream of the 3′ end, were assigned to the gene.

Using the MEME[55] curated Cis-BP[56] TF-binding motif reference, we scanned each ATAC-seq tumor type peak atlas and common atlas with FIMO[57] to find peaks likely to contain each motif ($P < 10^{-5}$). We filtered TFs that were not expressed in at least 50% of samples in at least one of the five tumor types. Further, similarity of predicted target peak sets was measured using the Jaccard index (size of intersection/size of union). If two TFs had a high Jaccard index (>0.5), we looked at the mean Jaccard index of each TF with all other TFs, and we removed the TF with the largest mean Jaccard index. The final set contained 352 motifs.

We created a matrix that defines a candidate set of associations between TFs and target genes: TF-binding site identification was used to turn each gene's set of assigned ATAC peaks into a feature vector of binding signals by assigning the maximum score of each motif across all peaks to a gene.

**Differential peak accessibility**. Reads aligning to atlas peak regions were counted using the countOverlaps function of the R packages GenomicAlignments and GenomicRanges[58]. Differential accessibility of these peaks was then calculated for all pairwise comparisons of cancer types using DESeq2[59].

**Motifs underlying differential accessibility in cell lines**. The shift in the cumulative distribution of chromatin accessibility changes (log$_2$-fold change) of the subset of the atlas occupied by each TF, compared to that of the background atlas, was measured by a one-sided KS test in either direction. The foreground occurrence is the number of peaks containing a particular TF motif within the group of 5000 differentially open or 5000 differentially closed peaks according to log$_2$-fold change read counts. The background occurrence is the number of peaks containing a particular TF motif among all the differentially accessible peaks.

**TCGA ATAC-seq analysis**. Currently, only bigwig files are publicly available for TCGA ATAC-seq but not raw data. We extracted the sum of ATAC-seq signals $\pm 0.5$ kb from the peak center for differentially accessible cell line peak regions for patients from these bigwig files and used these values for statistical analyses.

**Multitask learning**. For MTL, we trained regression models jointly for all tumors using *grouping and overlap in MTL* (GO-MTL)[17]. In this approach, prediction of each gene expression $\mathbf{y}_t$ is considered one task, and we wish to solve $T$ tasks jointly

so that information is "shared" between them. Let $\mathbf{X}$ be the data matrix of size $d \times N$ where each row represents a gene and each column is a motif hit score representing the target genes of a TF. We assume there are $K(<T)$ latent basis tasks and each observed task can be represented as linear combination of a subset of these basis tasks.

Briefly, we jointly learn regression models $\mathbf{w}_t$ that predict gene expression as linear combinations of latent regulatory programs in tumors. Formally, we learn a model matrix $\mathbf{W} = \mathbf{LS}$, where every column of matrix $\mathbf{W}$ is a model vector for a tumor transcriptional regulatory network, $\mathbf{w}_t$; $\mathbf{L}$ is matrix of latent tasks $\mathbf{L} = (\mathbf{l}_1 \dots \mathbf{l}_K)$; and $\mathbf{S}$ expresses the tumor transcriptional regulatory network models as linear combinations of the latent tasks, $\mathbf{w}_t = \mathbf{Ls}_t$. The model vector $\mathbf{w}_t$ represents the inferred global role of these TFs in driving gene expression; the event's true gene expression is denoted by $\mathbf{y}_t$; and the predicted gene expression is given by $\mathbf{w}_t \mathbf{X}_t$ (treating both as row vectors for notational convenience). The matrix $\mathbf{L}$ captures the predictive structure of the tasks and the grouping structure is determined by matrix $\mathbf{S}$. Tasks that have same sparsity pattern can be seen as belonging to the same group, while tasks whose sparsity patterns are orthogonal to each other can be seen as belonging to different groups. The partial sharing of latent basis tasks allows us to do away with the concept of disjoint groups. Any task that does not share latent bases with any other task in the pool can be seen as outlier task. Our learning cost function takes the following form:

$$\min_{\mathbf{L},\,\mathbf{S}} \sum_{t=1}^{T} \frac{1}{N} \left\| \mathbf{y}_t - \mathbf{X}'_t \mathbf{Ls}_t \right\|^2 + \mu \|\mathbf{S}\|_1 + \lambda \|\mathbf{L}\|_F^2 \qquad (1)$$

The parameter $\mu$ controls the sparsity in $\mathbf{S}$. The penalty on the Frobenius norm of $\mathbf{L}$ regularizes the predictor weights to have low $\ell_2$ norm and avoids overfitting.

To assess single task learning (STL) performance, we trained ridge regression models for each tumor (task) independently. We fit the ridge regression models using the SLEP MATLAB package and evaluated performance on held-out genes.

**In vitro drug-sensitivity analysis**. Detailed information on cell culture media is provided in the Supplementary Table 8. All cell lines were cultured under standard conditions at 37 °C and 5% CO$_2$. Cells were plated at 10–20% confluency (with the exception of JHUCS-1, MDA-MB-436, and RL-95 which were plated at ~50%) in 24-well plates in complete medium, and incubated inside an IncuCyte ZOOM system (Essen BioScience, Inc., MI, USA). The following day (22–24 h later), cells were exposed to LOR-253 (MedChemExpress, NJ, USA) at 0, 50, 250, or 1250 nM. To monitor cell growth, phase contrast images of the cell cultures in the presence or absence of the drug were captured automatically at 2-h intervals for up to 36 h, and occupied area of the cells (% confluency) was calculated using the IncuCyte image analysis software. We analyzed drug response data using a recently developed growth rate inhibition (GR) metric that corrects for differences in cell proliferation rates[60].

**Immunohistochemistry for ETV6 in UPSC**. The population considered for this study consisted of 31 patients diagnosed with uterine papillary serous carcinoma (UPSC) in stage III or IV, who underwent salpingo-oophorectomy at University of Texas MD Anderson Cancer Center (MDACC) and did not receive neoadjuvant therapy. Patients were divided in two groups based on survival: <2 years (eight patients) and more than 10 years (six patients). This study was approved by the Institutional Review Board at the MDACC. Informed consent was obtained from all patients. Formalin-fixed paraffin-embedded (FFPE) tumor blocks of archived UPSC were obtained from the repository of the Department of Gynecologic Oncology and Reproductive Medicine at MDACC. Clinical information was obtained from the electronic medical records.

FFPE 4 µm sections from patient tissues were deparaffinized and fixed in methanol prior to antigen retrieval in heated citrate buffer (pH 6.0, Poly Scientific R&D Corp.) at 120 °C for 7 min, followed by 10 min at 90 °C. Endogenous peroxidase was blocked with hydrogen peroxide solution (Millipore Sigma) 3% for 10 min. Protein blocking was performed using PBS-Tween 3%, BSA 1% donkey serum (Millipore Sigma) for 30 min. Anti-ETV6 antibody from Sigma (catalog number: HPA000264) at the titer of 1:500 was used. Samples were incubated with ETV6 for 1 h (1:75, polyclonal, Millipore Sigma) followed by use of MACH 3 rabbit HRP polymer detection (Biocare Medical). Antibodies were visualized by means of a dextran–polymer conjugate technique (EnVision+, Dako) using 3,3′-diaminobenzidine (DAB) (Dako) as chromogen. Tissue sections were counterstained with haematoxylin. Images were captured with a Leica DM LB microscope (Leica Microsistems).

Intensity of ETV6 stain was graded separately in nuclei and cytoplasm as 0 (negative), 1 (weak), 2 (moderate), and 3 (strong). Patient samples were divided into two groups based on this scoring: weak and medium nuclear (scores 0, 1 or 2) ETV6 staining, $n = 20$; and patient samples with strong (score = 3) nuclear ETV6 staining, $n = 11$. Statistical analysis studied the association between ETV6 staining intensity scores and survival time by using Kaplan–Meier curves and Log-rank test ($P < 0.004$).

**Immunohistochemistry for MITF in TNBC**. Immunohistochemical stain for MITF was performed on tissue microarrays (TMAs) containing triple negative breast carcinoma (TNBC). TNBC was defined as invasive breast carcinoma with

ER and PR staining in <1% of the tumor cells by immunohistochemistry and no HER2 overexpression by immunohistochemistry and no HER2 amplification by fluorescence in situ hybridization. Assessment of ER, PR, and HER2 follows the ASCO/CAP guidelines. Triplicate 0.6 mm diameter core from formalin-fixed paraffin-embedded TNBC blocks were used to construct the TMAs. MITF (D5) clone Dako Ab (catalog number: M3621) was used on Leica platform with ER2 pretreatment for 40 min. Pr Ab dilution: 1:50. Standard DAB Kit was used. MITF staining of any percentage and any intensity was considered positive. Some positivity is seen in the tumor-infiltrating lymphocytes. Clinical information was obtained from the electronic medical records. The association between MITF staining and survival was analyzed using Kaplan–Meier curves and log-rank test. This study was approved by the Institutional Review Board at MSKCC.

**Sample preparation for RNA-Seq and data analysis**. shRNA vector cloning: shRNA sequences for targeting human MITF were designed using the Splash algorithm prediction tool[61]. The shRNA was cloned into the LT3GEPIR miR-E backbone[37] enabling inducible shRNA expression in transduced cells upon doxycycline treatment. Two independent shRNA were used to target MITF (sh962 or shQa), and a previously described shRNA-targeting Renilla luciferase[62] was used as control. The sequences of all shRNAs can be found in Supplementary Table 9.

For protein lysates cells were incubated with RIPA buffer supplemented with protease inhibitors (Protease inhibitor tablets, Roche) for 30 min and cleared by centrifugation (15 min 14,000 rpms 4 C). Protein was quantified using the Bio-Rad protein assay (Cat. 500006). Primary antibody incubation was performed overnight at 4 °C in Tris-buffered saline containing 5% milk and 0.05% Tween-20. The following primary antibodies were used for immunoblotting: Mitf (ab12039, Abcam), Actin-HRP (A3854, Sigma). Mouse HRP-linked secondary antibody (GE Healthcare) was used and blots were developed with Lumi-Light Western Blotting Substrate (Roche).

MDA-MB-436 cells were maintained in RPMI 1640, supplemented with 10% FBS (Gemini, Cat. 900-208), 1X Glutamax (Gibco, Cat. 35050061), and penicillin–streptomycin (1%). MDA-MB-436 cells were transduced with a lentiviral construct (LT3GEPIR,[37]) and infected cells were then selected with puromycin (1 μg/ml, Sigma-Aldrich) and harvested at 5 or 17 days after culture in growth media containing doxycycline (1 μg/ml, Sigma-Aldrich). Total RNA was extracted using TRIzol (Thermal Fisher Scientific, Cat. 15596018), according to manufacturer's instructions. For RT-qPCR experiments, cDNA was obtained using Transcriptor First Strand cDNA Synthesis Kit (Roche, Cat. 04896866001). Gene-specific primer sets for human sequences were designed using PrimerBank [https://pga.mgh.harvard.edu/primerbank/] (see Supplementary Table 10 for qPCR primer sequences). *HPRT1* served as endogenous normalization controls. RT-qPCR was carried out in triplicate using PerfeCTa SYBR Green FastMix (QuantaBio, Cat. 95072-012) on the ViiA 7 Real-Time PCR System (Life technologies). For RNA-Seq, 500 μg of RNA was used, and PolyA mRNA was selected using beads coated with polyT oligonucleotides. Purified polyA mRNA was subsequently fragmented, and first and second strand cDNA synthesis performed using standard Illumina mRNA library preparation protocols (TruSeq RNA Sample Prep Kit v.2). Double-stranded cDNA was subsequently processed for TruSeq dual-index Illumina library generation. For sequencing ~30–40 million 80 bp single-end reads were acquired per replicate condition in a NextSeq Illumina system at the integrated genomics operation (IGO) Core at MSKCC.

Raw RNA-Seq reads were trimmed and filtered for quality using Trimmomatic[50]. Reads were aligned using STAR[63] against GRCh37.75(hg19). The RefSeq transcript annotations of the hg19 version of the human genome was used for the genomic location of transcription units. Genome-wide transcript counting was performed by HTSeq[64] to generate a matrix of raw counts. Differential expression of genes across cell types was calculated using DESeq2[59]. FDR threshold of 0.05 was imposed unless otherwise stated. A $\log_2$-fold change cutoff of 1 was used. We functionally annotated our expression profiling and performed gene set enrichment analysis[65] on all curated gene sets in the Molecular Signatures Database.

Pooled data is presented as mean ± standard deviation (SD) values of duplicate or triplicate biological replicates, as indicated in corresponding figure legends. Statistical significance differences compared to shMITF/shRen (for day 17) or shMITF +Dox/ −Dox (for day 5) were determined by an unpaired one-tailed Student's *t*-test. In figures, * stands for $P < 0.05$, ** for $P < 0.01$, and *** for $P < 0.001$.

**Reporting summary**. Further information on research design is available in the Nature Research Reporting Summary linked to this article.

## Data availability
The ATAC-seq and RNA-seq data have been deposited in the Gene Expression Omnibus accession number GSE114964 and GSE129337, respectively. The source data underlying Figs. 2, 3A–E, 4A–C, 5A, 6 and Supplementary Figs. 1, 2, 3, 6–10 are provided as Source Data files.

## Code availability
The software for PSIONIC is available from https://github.com/osmanbeyoglulab/PSIONIC.

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

## Acknowledgements

We would like to thank John I. Risinger for sending us the ACI158 and ACI126 cell lines, Sha Tian for her technical assistance in the RNA-seq library preparation, and Irina Linkov for performing MITF staining. The results published here are in whole or part based on data generated by The Cancer Genome Atlas project established by the NCI and NHGRI (accession number: phs000178.v7p6). Information about TCGA and the investigators and institutions that constitute the TCGA research network can be found at http://cancergenome.nih.gov/. This work was supported by NCI R21 award CA205819. H.U.O. is supported by NCI K99/R00 award CA207871.

## Author contributions

H.U.O and C.S.L. conceived and designed the study. H.U.O carried out the model training and computational validation. H.U.O and C.S.L. analyzed data and wrote the manuscript. F.S. performed the experimental validation for in vitro drug sensitivity and under supervision of G.C. and helped to write the experimental validation section. A.R.-V. and T.-L.Y. performed the ETV6 immunohistochemical staining under the supervision of S.C.M. and helped to write the experimental validation section. D.A.-C and H.-A.C performed MITF knock-down under supervision of S.W.L. and helped to write the experimental validation section. P.R. gathered clinical data for basal breast cancer patients. H.Y.W. analyzed MITF tissue microarrays. P.J. and D.A.L. assisted with the study design.

## Additional information

**Competing interests:** P.R. reports consulting/advisory board for Novartis and Iinstitutional Rresearch support from Illumina and GRAIL, Inc. The remaining authors declare no competing interests.

