## [Peer Review File · Nature Communications]

Reviewers' comments:

Reviewer #1 (Remarks to the Author):

The current manuscript develops a new computational method called PSIONIC (patient-specific inference of networks informed by chromatin) to exploit epigenomic data from relevant cancer models with expression data from patient tumours in order to identify the transcriptional programs and factors driving oncogenesis.

Overall, the idea of combining epigenomic data, such as chromatin accessibility, with gene expression to identify the disrupted transcription regulatory processes in cancers is very interesting. The multi-task learning approach that presented in the manuscript is a valid approach proposed by the authors. However, the advantages of this method over single-task learning are not obvious considering the performance analysis presented (Fig. 3A). Additionally, multitask learning does not seem to provide any advantage for retrieving patient specific inferred transcription factors activity. While the authors showcase regulatory differences across tumor types, they present limited validation and have not shown evidence that this method is applicable at the per-patient level.

Specific comments:

- The text needs major revisions. The paper generally reads more like the method section. Furthermore, parts of the text are repeated. Citations are missing in the introduction section and reference to figures does not always match. Figure legends (Fig1 for instance) are to be revised. Labels are also missing on a number of figure panels.
- Evaluation of performance of PSIONIC should be improved. The spearman correlation reported in Fig. 3A is considerably low. The authors did not mention how they selected the validation set of held-out genes nor provided any cross-validation or robustness of analysis. Considering the method, evaluation based on held-out tumors plus cross-validation would be more informative and needed.
- The authors provided one example of inferred transcription factors activity to be prognostic based on 14 samples in uterine serous tumors (Fig 4). Also MTF1 inferred activity is shown to be correlated with sensitivity to MTF1 inhibition in cell lines (Fig 5). However, these are fairly limited validation based on the number and/or analysis of the data presented. More evidence and validation in both analyses are recommended.
- The justification of selecting $K=7$ as the number of latent regulatory programs would need to be better justified.
- Inferred activities of TFs were linked to survival using cox proportional hazards regression model. While this analysis should be univariate, it is mentioned that the stage of the tumor was used as an additional covariate. Significant association of TFs with outcome has been claimed while no known TFs (ER, HER1 or cytokeratin based on this paper: <http://clincancerres.aacrjournals.org/content/10/16/5367.full-text.pdf>) for basal breast cancer are reported in the current manuscript. The authors should address this discrepancy.
- The authors mentioned training PSIONIC model on cell lines from CCLE. No validation and robustness for this analysis is reported. Also the concordance with the tumor inferred regulatory models need to be discussed in further details.
- MTF1 inferred activity is mentioned to be associated with growth rate (Fig 5) while no correlation is reported for this analysis.

Reviewer #2 (Remarks to the Author):

The manuscript describes the identification of TFs in tumorigenesis. Authors use ATAC-seq in tumor cell lines to map changes in accessibility indicative of differences in TF binding and relate this information to gene expression measured by RNA-seq in primary tumors using a novel computational model. Using this approach, authors confirm some previous findings and identify new TFs associated with patient survival. These factors could be used as therapeutic targets, and authors attempt to prove this by showing that inhibition of MTF1 decreases growth rates in tumor cell lines in which this factor is expressed at higher levels.

The results presented here, including computational strategies, are interesting and could potentially give important insights into personalized cancer treatment. Ideally, as described below, the ATAC-seq experiments should have been done using cells from the original tumors, but I realize that this may be difficult to accomplish without access to the original samples. The following are mostly minor comments asking for clarification of some important points:

1. Authors used ATAC-seq information obtained in cell lines to compare to RNA-seq obtained from patient samples. Cell lines have been selected for growth in culture, and they may be expressing transcription factors not present in the original tumors. What is known about these cell lines and how well they compare to the original tumors? Authors should comment on the appropriateness of this comparison. Is it expected that ovarian and breast cancer cell lines have similar ATAC-seq profiles?
2. Distances in kb and scales for the different tracks are missing in Figure 2B
3. Page 7. Authors should comment on whether differences in TF signal between cell lines is due to the tumor type or the tissue of origin. Are HNF1 motifs more accessible in normal endometrial cells?
4. Page 11, bottom. What is known about the role of MTF1 and how does this TF control cell growth?

Response to Reviewers' Comments

We have comprehensively addressed all comments from the reviewers, and we believe that we have greatly improved the clarity and organization of the manuscript.

Reviewer #1

The current manuscript develops a new computational method called PSIONIC (patient-specific inference of networks informed by chromatin) to exploit epigenomic data from relevant cancer models with expression data from patient tumours in order to identify the transcriptional programs and factors driving oncogenesis. Overall, the idea of combining epigenomic data, such as chromatin accessibility, with gene expression to identify the disrupted transcription regulatory processes in cancers is very interesting.

We appreciate that the reviewer finds our idea very interesting.

The multi-task learning approach that presented in the manuscript is a valid approach proposed by the authors. However, the advantages of this method over single-task learning are not obvious considering the performance analysis presented (Fig. 3A). Additionally, multitask learning does not seem to provide any advantage for retrieving patient specific inferred transcription factors activity.

MTL allows us to make direct comparisons between patient specific models. As can be seen from **Fig. R1**, the t-SNE plot of inferred TF activities mostly separates patients according to their tumor types, while this is not true when clustering models from the single task learning (STL) approach. Using STL, comparing different models is not straightforward.

As can be seen from **Fig. R2**, we obtained significantly better regression performance with our MTL approach, PSIONIC, compared to a STL approach based on ridge regression ($P < 10^{-21}$, one-sided Wilcoxon signed-rank test).

Fig. R1: t-SNE projections of inferred TF activities for **(A)** PSIONIC model and **(B)** STL model.

While the authors showcase regulatory differences across tumor types, they present limited validation and have not shown evidence that this method is applicable at the per-patient level.

We added the following new data and validation experiments to address this comment:

- (1) We validated tumor-type specific accessibility signatures derived from our cell line models using TCGA ATAC-seq for UCEC-ENDO and BRCA-BASAL (**Supplementary Figure 1**).
- (2) We collected additional ETV6 immunostaining data in serous endometrial tumor samples and updated the ETV6 survival figure (**Fig. 5A**) with increased sample size.
- (3) We performed new immunohistochemical analyses in primary tumor samples from patients with basal breast cancer for MITF to validate prognostic TF markers identified by PSIONIC (**Fig. 6A-B**).
- (4) We manipulated MITF protein levels by shRNA-mediated knockdown in a basal breast cell line (MDA-MB-436) and performed RNA-seq to further validate role of MITF in basal breast cancer (**Fig. 6C-E, Supplementary Figure 9**), in particular, showing consistent MITF regulation of target genes in the experimental data with those whose expression levels correlate with PSIONIC-inferred activity across cell lines or tumors.

In particular, the survival data and consistency of target gene expression patterns suggest that patient-specific inferred activities indeed provide prognostic and gene regulatory information.

Specific comments:

- The text needs major revisions. The paper generally reads more like the method section. Furthermore, parts of the text are repeated. Citations are missing in the introduction section and reference to figures does not always match. Figure legends (Fig1 for instance) are to be revised. Labels are also missing on a number of figure panels.

We have performed substantial text revisions and moved some technical details to the **Methods**. We also added citations in the introduction section and updated references to figures. We apologize for previous missing elements in the figure legends; we have revised figure legends and added missing labels of figure panels.

- Evaluation of performance of PSIONIC should be improved. The spearman correlation reported in Fig. 3A is considerably low.

As the reviewer correctly observes, when we predict held-out tumor gene expression from gene regulatory sequence derived from cell line ATAC-seq data, the mean Spearman rank correlation between predicted and measured gene expression is modest (mean $\rho = 0.384$, see **Fig. R2**). Note, however, that this is a very hard problem: tumor samples are heterogenous, adding to noise in the log fold changes in gene expression that we are trying to predict; patient-specific chromatin accessibility data are not available for our full cohort, so we use high quality cell line ATAC-seq data as a proxy; true enhancer-gene associations are unknown. Nevertheless, we can use the model to derive meaningful TF activities that predict drug response and survival, as shown in our results. To improve our evaluation, we have added 10-fold cross validation analysis for TCGA and CCLE models (**Fig. R2 and Fig. R4**).

The authors did not mention how they selected the validation set of held-out genes nor provided any cross-validation or robustness of analysis. Considering the method, evaluation based on held-out tumors plus cross-validation would be more informative and needed.

We performed 10-fold cross validation analysis and updated **Fig. 3A**. As shown in **Fig. R2**, in 10-fold cross-validation experiments on held-out genes, we obtained a mean Spearman rank correlation between predicted and measured gene expression changes of 0.384 ± 0.016 , a modest but highly significant result ($P < 10^{-16}$, one-sided Wilcoxon signed-rank test). Further, we obtained significantly better regression performance than a STL approach based on ridge regression ($P < 10^{-21}$, one-sided Wilcoxon signed-rank test). Similarly, our models with motif data from promoter and enhancer regions outperformed models where only motif hits in promoter regions were used ($P < 10^{-16}$, one-sided Wilcoxon signed-rank test). In contrast, if we randomized motif hits for each chromatin accessible region across all motifs, or if we randomized accessible regions for each motif, then assigned to the nearest gene, the prediction performance also significantly decreased ($P < 10^{-32}$, one-sided Wilcoxon signed-rank test).

Fig. R1 (Fig. 3A): PSIONIC and STL regression models predict differential expression of held-out genes and subtypes of tumor samples. Plot showing Spearman correlations between predicted and actual gene expression changes for all samples, sorted based on performance of the PSIONIC model using enhancer and promoter TF binding sites. For each method and each sample, the Spearman correlation is computed using 10-fold cross-validation on held-out genes. Using TF binding sites from enhancer promoter as features (mean $\rho = 0.384 \pm 0.016$) is significantly better than we randomized motif hits for each chromatin accessible region across all motifs (mean $\rho = 0.144 \pm 0.022$; $P < 10^{-32}$, one-sided Wilcoxon signed-rank test), or if we randomized accessible regions for each motif, then assigned to the nearest gene (mean $\rho = 0.235 \pm 0.025$; $P < 10^{-32}$, one-sided Wilcoxon signed-rank test). PSIONIC models with motif data from promoter and enhancer regions outperformed models where only motif hits in promoter regions were used (mean $\rho = 0.337 \pm 0.012$; $P < 10^{-16}$, one-sided Wilcoxon signed-rank test) and STL approach based on ridge regression (mean $\rho = 0.352 \pm 0.019$; $P < 10^{-21}$, one-sided Wilcoxon signed-rank test). TCGA tumor types are shown in the top bar.

We could not perform cross-validation analysis based on held-out tumors since our current computational framework does not allow us held-out tumors completely. That is, while MTL learns patient-specific models that are comparable to each other, we cannot learn a model for a held-out patient (i.e. without expression data for at least some genes for the patient).

- The authors provided one example of inferred transcription factors activity to be prognostic based on 14 samples in uterine serous tumors (Fig 4). Also, MTF1 inferred activity is shown to be correlated with sensitivity to MTF1 inhibition in cell lines (Fig 5). However, these are fairly limited validation based on the number and/or analysis of the data presented. More evidence and validation in both analyses are recommended.

Although uterine serous is a rare cancer type, we were able to perform additional ETV6 staining in uterine serous tumors to increase our cohort size. We updated the ETV6 survival figure (**Fig. 4A**) with increased sample size. For the current analysis, 31 patients with stage III or IV UPSC were studied as 2 groups (patients with cytoplasmic, negative or weak / medium [score=1 or 2] nuclear ETV6 staining, N=20; and patients with strong [score=3] nuclear ETV6 staining, N=11) by Kaplan-Meier analysis and log-rank test. When we increased the sample size, we got

statistically more significant results (cytoplasmic or weak nuclear ETV6: median survival of 2330 days (95% CI: 104-4556 days); stronger nuclear ETV6: median survival of 214 days (95% CI: 53-891); log-rank $P = 0.004$).

- The justification of selecting $K=7$ as the number of latent regulatory programs would need to be better justified.

When we compare 10-fold cross validation results with different values of K , we found that prediction performance was stable after $K = 4$, with no sign of overfitting with higher K . However, a higher number of regulatory programs does allow PSIONIC-inferred models to better distinguish between tumors of distinct subtypes (**Fig. R3, Supplementary Figure 4**). Therefore, $K = 7$ seems like a reasonable choice for optimizing both overall prediction performance and capturing tumor-type specific components of the regulatory models. We have added this justification to the text.

Fig. R2 (Supplementary Figure 4): t-SNE projections of inferred TF activities with varying K values. 10-fold cross validation results, reported as mean Spearman correlation values, are: K = 2: 0.383 ± 0.019 ; K = 3: 0.383 ± 0.016 ; K = 4: 0.384 ± 0.016 ; K = 5: 0.384 ± 0.016 ; K = 6: 0.384 ± 0.016 ; K = 7: 0.384 ± 0.016 ; K = 8: 0.384 ± 0.016 ; K = 9: 0.384 ± 0.016 .

- Inferred activities of TFs were linked to survival using cox proportional hazards regression model. While this analysis should be univariate, it is mentioned that the stage of the tumor was used as an additional covariate.

We added clinical stage and age as additional covariates in order to show inferred activities of TFs associated with survival after controlling for these factors. We have clarified this in the text.

Significant association of TFs with outcome has been claimed while no known TFs (ER, HER1 or cytokeratin based on this paper: <http://clincancerres.aacrjournals.org/content/10/16/5367.full-text.pdf>) for basal breast cancer are reported in the current manuscript. The authors should address this discrepancy.

Previous immunohistochemical analyses of HOXB9¹ and ZEB1² showed that the expression of these TFs was significantly associated with prognosis of triple-negative breast cancer patients, consistent with our result. In clinical breast cancer specimens, Zhang et al. showed that SOX4 was abnormally overexpressed and correlated with the triple-negative breast cancer subtype (ER-/PR-/HER2-)³.

We have performed new immunohistochemical analyses in primary tumor samples from MSKCC patients with basal breast cancer (n = 45) for MITF to validate the localization of this PSIONIC-identified prognostic TF marker. MITF inferred activity separated patients into high- and low-risk groups in BASAL-BRCA (FDR = 0.011, Cox analysis). Indeed, tissue microarray analyses in clinically-annotated primary basal breast tumor samples (n = 45) validated MITF positivity in tumor cells and revealed a significant association between MITF expression and patient survival (log-rank test, $P < 0.006$), with median survival of 1208 and 2406 days for the positive and negative staining groups, respectively (see Kaplan-Meier survival curve and representative MITF positive staining in basal breast cancer patients in **Fig. R4-6A, B**). MITF is a key TF in melanocyte development and differentiation and a diagnostic biomarker for metastatic melanoma⁴. However, the role of MITF in non-melanoma cancer cells, including basal breast cancer, is largely undefined. Thus, we next sought to functionally validate PSIONIC-predicted MITF activity in basal breast cancer cells.

To this end, we generated inducible shRNA vectors⁵ targeting MITF and evaluated their impact on basal breast cancer gene expression. Potent shRNA-driven MITF downregulation was confirmed in both MDA-MB-436 basal breast cancer cells and SK-Mel-28 melanoma cells with known high MITF levels (**Supplementary Figure 9A-C**). RNA-seq following MITF silencing revealed an effect on gene expression with 58 consistently down-regulated and 103 consistently up-regulated genes (adjusted P -value < 0.05 and fold change > 2) in MDA-MB-436 cells transduced with two independent MITF shRNAs (**Fig. 6C; Supplementary Table 6**). Interestingly, commonly down-regulated genes included c-Myc and c-Myc target genes, as well as additional pro-oncogenic factors such as IL1B, NT5E (CD73) and other molecules with functions in tumor immune escape (**Fig. R4-6D, Supplementary Table 7-8**)^{6,7}, which were validated by qPCR (**Fig. S9C**). Commonly upregulated genes were enriched in ontology terms associated with immune activation (defensins, complement, IFN, IL15, CCL2) and cell adhesion (e.g. SVEP1) (**Fig. R4-6D, Supplementary Table 7-8, Supplementary Figure 9D**). These effects were not associated with changes in the proliferation rate of MDA-MB-436 cells *in vitro* (not shown) yet are suggestive of an *in vivo* role for MITF in the regulation of cancer – microenvironment crosstalk in basal breast cancer. Importantly, most differentially expressed genes (DEGs) identified in MDA-MB-436 upon MITF suppression correlated with PSIONIC-

inferred MITF activity across multiple basal breast cancer cell lines (n = 29; 75 out of 161 DEG, ~47%, $|\rho| > 0.4$, **Fig. 6E**) as well as across patient samples (n = 92; 43 out of 161 DEG, ~27%, $|\rho| > 0.4$). Together, these results validate the prediction made by PSIONIC on MITF activity and gene regulation in basal breast cancer.

Fig. R4 (Fig. 6). Clinical and in vitro validation of MITF in basal breast cancer.

(A) Kaplan-Meier plot for basal breast cancer patients stratified by MITF staining score. Patient samples (n = 45) were divided into two groups based on MITF positivity (n=10) and negativity (n= 35) in nuclei or cytoplasm staining. A significant difference in survival was observed between the groups (log-rank test, $P = 0.006$). The median survival was 1208 days for positive staining group and 2406 days for negative staining group. **(B)** Representative image of IHC staining with MITF antibody on a primary basal breast cancer tumor. **(C)** Volcano plot depicting the changes in representation (\log_2 fold change; x-axis) and significance ($-\log_{10}$ adjusted P -value; y-axis) of *Mitf* shRNA versus dox-inducible *Ren* expressing MDA-MB-436 cells at day 17. **(D)** Hallmarks of cancer and REACTOME gene sets analyzed from the transcriptome analysis comparing MDA-MB-436 cells transduced with two independent MITF shRNAs and control. Enrichment score (ES) is shown. **(E)** RNA-seq expression (row-normalized) for the subset of differentially expressed genes where gene expression correlated with PSIONIC-inferred MITF activity across breast cancer cell lines target genes with $|\rho| > 0.4$ shown). Red labels indicate positive correlation, blue labels indicate negative correlation. Bold labels indicate the existence of correlation in TCGA BASAL-BRCA tumors.

- The authors mentioned training PSIONIC model on cell lines from CCLE. No validation and robustness for this analysis is reported.

We have added 10-fold cross validation analysis for models on cell lines from CCLE in **Supplementary Figure 6**. As shown in the **Fig. R5**, in 10-fold cross-validation experiments on held-out genes, we obtained a mean Spearman rank correlation between predicted and measured gene expression changes of 0.387, a modest but highly significant result relative to baseline methods. In particular, we obtained significantly better regression performance than a STL approach based on ridge regression ($P < 10^{-21}$, one-sided Wilcoxon signed-rank test). Similarly, our models with motif data from promoter and enhancer regions outperformed models where only motif hits in promoter regions were used ($P < 10^{-16}$, one-sided Wilcoxon signed-rank test). In contrast, if we randomized motif hits for each chromatin accessible region across all motifs, or if we randomized accessible regions for each motif, then assigned to the nearest gene, the prediction performance also significantly decreased ($P < 10^{-32}$, one-sided Wilcoxon signed-rank test).

Fig. R5 (Supplementary Figure 6): PSIONIC and STL regression models predict differential expression of held - out genes and subtypes of cell lines from CCLE. Plot showing Spearman correlations between predicted and actual gene expression changes for all samples, sorted based on performance of the PSIONIC model using enhancer and promoter TF binding sites. For each method and each cell line, the Spearman correlation is computed using 10-fold cross-validation on held-out genes. Using TF binding sites from enhancer promoter as features (mean $\rho = 0.387 \pm 0.018$) is significantly better than we randomized motif hits for each chromatin accessible region across all motifs (mean $\rho = 0.116 \pm 0.007$; $P < 10^{-32}$, one-sided Wilcoxon signed-rank test), or if we randomized accessible regions for each motif, then assigned to the nearest gene (mean $\rho = 0.128 \pm 0.014$; $P < 10^{-32}$, one-sided Wilcoxon signed-rank test). PSIONIC models with motif data from promoter and enhancer regions outperformed models where only motif hits in promoter regions were used (mean $\rho = 0.364 \pm 0.016$; $P < 10^{-16}$, one-sided Wilcoxon signed-rank test) and STL approach based on ridge regression (mean $\rho = 0.352 \pm 0.021$; $P < 10^{-21}$, one-sided Wilcoxon signed-rank test). TCGA tumor types are shown in the top bar.

Also the concordance with the tumor inferred regulatory models need to be discussed in further details.

We mean centered tumor and cell line inferred transcription factor activities (W) and we use t-SNE to cluster profile specific regulatory models. As can be seen from **Supplementary Figure**

5/ Fig. R6, regulatory models for cell lines to some extent recapitulated patient-specific tumor regulatory models (**Supplementary Figure S5**). Importantly, cell line models as well as tumor models were clustered mostly within themselves. For example, ovarian cell line models as indicated in orange plus symbols clustered mostly with OV tumors.

Fig. R6 (Supplementary Figure 5): t-SNE projections of mean centered inferred TF activities for TCGA patients (as denoted with squares) and CCLE cell lines (as denoted with plus sign).

- MTF1 inferred activity is mentioned to be associated with growth rate (Fig 5) while no correlation is reported for this analysis.

We had planned to include this correlation value in the original supplementary document and apologize for the oversight. We updated the text as below as well as **Fig. 5**: “MTF1 inferred activity was significantly associated with growth rate inhibition by Spearman correlation analysis ($P < 10^{-2}$ for these cell lines, $\rho = 0.795$).”

Reviewer #2

The manuscript describes the identification of TFs in tumorigenesis. Authors use ATAC-seq in tumor cell lines to map changes in accessibility indicative of differences in TF binding and relate this information to gene expression measured by RNA-seq in primary

tumors using a novel computational model. Using this approach, authors confirm some previous findings and identify new TFs associated with patient survival. These factors could be used as therapeutic targets, and authors attempt to prove this by showing that inhibition of MTF1 decreases growth rates in tumor cell lines in which this factor is expressed at higher levels. The results presented here, including computational strategies, are interesting and could potentially give important insights into personalized cancer treatment.

We appreciate that the reviewer finds our computational strategy and results interesting.

Ideally, as described below, the ATAC-seq experiments should have been done using cells from the original tumors, but I realize that this may be difficult to accomplish without access to the original samples. The following are mostly minor comments asking for clarification of some important points:

1. Authors used ATAC-seq information obtained in cell lines to compare to RNA-seq obtained from patient samples. Cell lines have been selected for growth in culture, and they may be expressing transcription factors not present in the original tumors.

What is known about these cell lines and how well they compare to the original tumors? Authors should comment on the appropriateness of this comparison.

In this study, we chose cell lines widely used as representative of corresponding tumor types depending on availability to our group. In several cases, we are providing the first epigenomic characterization of these cell line models.

Endometrial endometrioid cell lines. Ishikawa and RL-95-2 derived from type I and KLE and AN3CA derived from type II endometrial carcinomas tumors have been widely used as models to investigate molecular genetics and mechanisms underlying their development, progression and response to therapeutics⁸. KLE and AN3CA cells, originating from peritoneal and lymph node metastases, respectively, and RL-95-2 cells derived from a moderately differentiated (Grade 2) endometrial adenosquamous carcinoma. Ishikawa cells were established from the epithelial component of a moderately differentiated, stage 2, endometrial adenocarcinoma.

Ovarian cell lines. CAOV3 and OVCAR8 have been widely used as representatives of high-grade serous cancer. CAOV3 and OVCAR8 possess *TP53* mutations and substantial copy-number changes, key characteristics of high grade serous ovarian cancer (HGSOC).

Uterine serous cell lines. ARK1, ARK2, ACI-158 and ACI-126 are the main uterine serous (UPSC) cell lines. However, they are not commercially available. Dr. John I. Risinger kindly sent us the ACI158 and ACI126 cell lines. Hence, we chose these two cell lines for uterine serous cancer.

Uterine carcinosarcoma cell lines. JHUCS-1 and SNU685 are uterine carcinoma cell lines available to our group. JHUCS-1 was established from a carcinosarcoma (malignant mixed mesodermal tumor) of the uterus that was surgically removed from a 57-year-old Japanese woman⁹. SNU-685 was derived from uterine malignant mixed mullerian tumor¹⁰.

In order to investigate how well these cell lines compare to primary tumors, we used recently published ATAC-seq profiles of tumor samples from TCGA¹¹ including 13 UCEC-ENDO (24 with replicates) and 15 BRCA-BASAL (30 with replicates). First, we examined patterns of gain or

loss of chromatin accessible regions between endometrial and basal breast cancer cell lines ($FDR < 10^{-4}$, $\log_2(FC) > 3$). We identified 368 peaks that were more accessible in basal breast cancer cell lines and 366 peaks more accessible in endometrial cell lines. Currently only bigwig files are publicly available for TCGA ATAC-seq but not raw data. We extracted the sum of ATAC-seq signals ± 0.5 kb from peak center for differentially accessible cell line peak regions for patients from these bigwig files. Consistent with our cell line data, high accessibility regions in breast cancer cell lines displayed significantly higher accessibility in BRCA-BASAL patients compared to UCEC-ENDO ($P < 10^{-4}$, one-sided Wilcoxon signed-rank test). Similarly, high accessibility regions in uterine endometrioid cell lines showed significantly higher accessibility in UCEC-ENDO patients compared to BRCA-BASAL ($P = 0.00016$, one-sided Wilcoxon signed-rank test) as shown in **Fig. R7**.

Fig. R7 (Supplementary Figure 1): Comparison of ATAC-seq profiles of BRCA-BASAL and UCEC-ENDO tumors in differential chromatin accessibility regions from basal breast and endometrial cell lines: **(A)** BRCA-BASAL tumors have significantly higher sum of ATAC-seq signal compared to UCEC-ENDO tumors at loci with increased chromatin accessibility in basal breast cancer cell lines compared to endometrial cell lines ($P < 10^{-4}$, one-sided Wilcoxon signed-rank test); **(B)** UCEC-ENDO tumors have significantly higher sum of ATAC-seq signal compared to BRCA-BASAL tumors at loci with increased

chromatin accessibility in endometrial cell lines compared to basal breast cancer cell lines ($P = 0.00016$, one-sided Wilcoxon signed-rank test)

Is it expected that ovarian and breast cancer cell lines have similar ATAC-seq profiles?

The TCGA Research Network uncovered notable genomic similarities between the basal-like breast cancer subtype and serous ovarian cancer. The mutation spectrum, or types and frequencies of genomic mutations, were largely the same in both cancer types (e.g. widespread copy number alterations and frequent mutations in TP53 gene).

2. Distances in kb and scales for the different tracks are missing in Figure 2B

We updated **Fig. 2B**. All y-axis scales now range from 0–235 in normalized arbitrary units. The x-axis scale is indicated by the scale bars.

3. Page 7. Authors should comment on whether differences in TF signal between cell lines is due to the tumor type or the tissue of origin.

In some cases the TF signal between cell lines might indeed be due to the tissue of origin. To look more closely at this issue, we examined publicly available chromatin accessibility data in relevant normal tissues.

We generated a reference chromatin accessibility atlas for normal uterine (n=1), ovarian (n=3), and breast (n=1) tissue using DNase-seq data by the Roadmap Epigenomics project¹² and assembled an atlas of ~397K accessibility regions. We performed motif analysis in each chromatin accessible regions in the common atlas. Then, we examined the patterns of gain or loss of chromatin accessible regions between each pair of tumor types by performing pairwise differential read count analysis on accessible regions.

Fig. R8: Pairwise comparison of transcription factor motifs enriched in differentially accessible regions in normal tissues Uterus vs Ovary. Volcano plot showing effect size versus $-\log_{10}(P)$, using a Bonferroni correction to adjust P values for each plot. TF symbol annotations are written where the adjusted

$P < 10^5$. The foreground occurrence is the number of peaks containing a particular TF motif within the group of differential accessible peaks according to log2 fold-change read counts, respectively. The background occurrence is the number of peaks containing a particular TF motif found among all the differentially accessible peaks.

Are HNF1 motifs more accessible in normal endometrial cells?

HNF1 motifs are not more accessible in normal uterus cells from the Roadmap Epigenomics project when we performed pairwise analysis. We also performed motif analysis on 144,535 peak regions from these uterus tissue samples; interestingly, only 329 peak regions contain the HIF1A motif and 291 regions contain HIF1B motif.

4. Page 11, bottom. What is known about the role of MTF1 and how does this TF control cell growth?

The metal response element-binding transcription factor-1 (MTF-1) is a ubiquitously expressed transcription factor that is activated by heavy metals, redox stresses, growth factors and cytokines¹³. Under normal conditions, MTF1 localizes both to the nucleus and the cytoplasm but accumulates in the nucleus upon these diverse stresses. After binding DNA, MTF1 recruits different co-regulators and often relies on other transcription factors such as p300 /CBP, Sp1, and HIF1 α for coordinated target gene expression. Its established targets have important roles in metal homeostasis, embryonic development, tumor progression, and oxidative stress or hypoxia signaling. Importantly, inhibition of MTF1 induces the expression of tumor suppressor factor Kruppel like factor 4 (KLF4)¹⁴. This leads to the downregulation of cyclin D1, blocking cell cycle progression and proliferation. MTF1 inhibitor LOR-253 enhances apoptosis induced by cisplatin in both SKOV3 and OVCAR3 cells¹⁵, is cytotoxic to Raji and Raji/253R lymphoma cell lines¹⁶, and suppresses the proliferation of acute myeloid leukemia (AML) cell lines¹⁷. A clinical trial testing LOR-253 in patients with AML and myelodysplastic syndrome is currently ongoing (ClinicalTrials.gov: NCT02267863).

References

- 1 Seki, H. *et al.* HOXB9 expression promoting tumor cell proliferation and angiogenesis is associated with clinical outcomes in breast cancer patients. *Ann Surg Oncol* **19**, 1831-1840, doi:10.1245/s10434-012-2295-5 (2012).
- 2 Jang, M. H., Kim, H. J., Kim, E. J., Chung, Y. R. & Park, S. Y. Expression of epithelial-mesenchymal transition-related markers in triple-negative breast cancer: ZEB1 as a potential biomarker for poor clinical outcome. *Hum Pathol* **46**, 1267-1274, doi:10.1016/j.humpath.2015.05.010 (2015).
- 3 Zhang, J. *et al.* SOX4 induces epithelial-mesenchymal transition and contributes to breast cancer progression. *Cancer research* **72**, 4597-4608, doi:10.1158/0008-5472.CAN-12-1045 (2012).
- 4 Hartman, M. L. & Czyz, M. MITF in melanoma: mechanisms behind its expression and activity. *Cell Mol Life Sci* **72**, 1249-1260, doi:10.1007/s00018-014-1791-0 (2015).

- 5 Fellmann, C. *et al.* An optimized microRNA backbone for effective single-copy RNAi. *Cell Rep* **5**, 1704-1713, doi:10.1016/j.celrep.2013.11.020 (2013).
- 6 Buisseret, L. *et al.* Clinical significance of CD73 in triple-negative breast cancer: multiplex analysis of a phase III clinical trial. *Ann Oncol* **29**, 1056-1062, doi:10.1093/annonc/mdx730 (2018).
- 7 Loi, S. *et al.* CD73 promotes anthracycline resistance and poor prognosis in triple negative breast cancer. *Proc Natl Acad Sci U S A* **110**, 11091-11096, doi:10.1073/pnas.1222251110 (2013).
- 8 Korch, C. *et al.* DNA profiling analysis of endometrial and ovarian cell lines reveals misidentification, redundancy and contamination. *Gynecol Oncol* **127**, 241-248, doi:10.1016/j.ygyno.2012.06.017 (2012).
- 9 Yamada, K. *et al.* Establishment and characterization of JHUCS-1 cell line derived from carcinosarcoma of the human uterus. *Hum Cell* **17**, 139-144 (2004).
- 10 Yuan, Y. *et al.* Establishment and characterization of cell lines derived from uterine malignant mixed Mullerian tumor. *Gynecol Oncol* **66**, 464-474, doi:10.1006/gyno.1997.4802 (1997).
- 11 Corces, M. R. *et al.* The chromatin accessibility landscape of primary human cancers. *Science* **362**, doi:10.1126/science.aav1898 (2018).
- 12 Roadmap Epigenomics, C. *et al.* Integrative analysis of 111 reference human epigenomes. *Nature* **518**, 317-330, doi:10.1038/nature14248 (2015).
- 13 Gunther, V., Lindert, U. & Schaffner, W. The taste of heavy metals: gene regulation by MTF-1. *Biochimica et biophysica acta* **1823**, 1416-1425, doi:10.1016/j.bbamcr.2012.01.005 (2012).
- 14 Kindermann, B., Doring, F., Budczies, J. & Daniel, H. Zinc-sensitive genes as potential new target genes of the metal transcription factor-1 (MTF-1). *Biochem Cell Biol* **83**, 221-229, doi:10.1139/o04-133 (2005).
- 15 Wang, B. *et al.* KLF4 expression enhances the efficacy of chemotherapy drugs in ovarian cancer cells. *Biochem Biophys Res Commun* **484**, 486-492, doi:10.1016/j.bbrc.2017.01.062 (2017).
- 16 Tsai, C. Y. *et al.* APTO-253 Is a New Addition to the Repertoire of Drugs that Can Exploit DNA BRCA1/2 Deficiency. *Mol Cancer Ther* **17**, 1167-1176, doi:10.1158/1535-7163.MCT-17-0834 (2018).
- 17 Local, A. *et al.* APTO-253 Stabilizes G-quadruplex DNA, Inhibits MYC Expression, and Induces DNA Damage in Acute Myeloid Leukemia Cells. *Mol Cancer Ther* **17**, 1177-1186, doi:10.1158/1535-7163.MCT-17-1209 (2018).

REVIEWERS' COMMENTS:

Reviewer #1 (Remarks to the Author):

The authors have addressed my concerns.

Reviewer #2 (Remarks to the Author):

I only had minor comments asking authors to clarify some points I felt there were important for the non-expert to understand the manuscript. The authors have addressed all these points in the response to the reviewers. However, it appears that authors did not integrate this information in the manuscript. For example, I couldn't find a place in the manuscript where the authors briefly describe the characteristics and properties of the cell lines, so that the reader can understand the significance of the results obtained with these cell lines in the context of different tumor types. Similarly, I couldn't find where the authors explained in the revised manuscript if the differences in TF signal between cell lines is due to the tumor type or the tissue of origin. I apologize if it is in the manuscript and I didn't see it. Other than this, I think the manuscript is appropriate for publication in Nat Comm

Reviewer #3 (Remarks to the Author):

The manuscript presents a new computational method, PSIONIC, that combines chromatin accessibility with gene expression data to better understand the effect of enhancers in transcriptional programs in cancer. The idea of leveraging epigenomic data, as well as the use of Multi task learning, is novel and the results presented in the article are promising. The authors go one step further and experimentally validate the ability of two PSIONIC predicted transcription factors (ETV6 and MITF) to influence prognostic outcome. This article has the potential of influencing personalized cancer therapy. The authors performed extensive validation studies in response to the first round of reviews. I find that they have adequately addressed the critiques and therefore I recommend this manuscript for publication.

REVIEWERS' COMMENTS:

Reviewer #1 (Remarks to the Author):

The authors have addressed my concerns.

Reviewer #2 (Remarks to the Author):

I only had minor comments asking authors to clarify some points I felt there were important for the non-expert to understand the manuscript. The authors have addressed all these points in the response to the reviewers. However, it appears that authors did not integrate this information in the manuscript.

For example, I couldn't find a place in the manuscript where the authors briefly describe the characteristics and properties of the cell lines, so that the reader can understand the significance of the results obtained with these cell lines in the context of different tumor types.

We added a section on *Cell line selection for ATAC-seq* to the **Methods**. For convenience, we repeat the new text below:

Cell line selection for ATAC-seq

In this study, we chose cell lines widely used as representative of corresponding tumor types depending on availability to our group. In several cases, we are providing the first epigenomic characterization of these cell line models. ATAC-seq libraries generated from basal breast (MDA-MB-231, MDA-MB-436) high-grade serous ovarian (OVCAR8, Caov3), uterine carcinosarcoma (JHUCS, SNU685), endometrial endometrioid (AN3-CA, KLE, Ishikawa, RL95-2) and serous carcinoma (ACI-126, ACI-158) cell lines. Gynecologic cell lines OVCAR8, Caov3, JHUCS, SNU685, AN3-CA, KLE, Ishikawa, and RL95-2 were supplied by Douglas A. Levine. Uterine serous cell lines ACI-126 and ACI-158 were kindly supplied by John I. Risinger from Michigan State University. Basal breast cancer cell lines MDA-MB-231 and MDA-MB-436 were acquired from ATCC. The cell lines have been tested negative for mycoplasma contamination.

Briefly, Ishikawa and RL-95-2 derived from type I and KLE and AN3CA derived from type II endometrial carcinomas tumors have been widely used as models to investigate molecular genetics and mechanisms underlying their development, progression and response to therapeutics¹. KLE and AN3CA cells, originating from peritoneal and lymph node metastases, respectively, and RL-95-2 cells derived from a moderately differentiated (Grade 2) endometrial adenosquamous carcinoma. Ishikawa cells were established from the epithelial component of a moderately differentiated, stage 2, endometrial adenocarcinoma. CAO3 and OVCAR8 have been widely used as representatives of high-grade serous cancer. CAO3 and OVCAR8 possess *TP53* mutations and substantial copy-number changes, key characteristics of high grade serous ovarian cancer (HGSOC). ACI-158 and ACI-126 are the main uterine serous (UPSC) cell lines. JHUCS-1 was established from a carcinosarcoma (malignant mixed mesodermal tumor) of the uterus that was surgically removed from a 57-year-old Japanese woman². SNU-685 was derived from uterine malignant mixed mullerian tumor³.

Similarly, I couldn't find where the authors explained in the revised manuscript if the differences in TF signal between cell lines is due to the tumor type or the tissue of origin.

We tried to clarify this point by adding the section below to **Results**:

In some cases the TF signal between cell lines might be due to the tissue of origin. To look more closely at this issue, we examined publicly available chromatin accessibility data in relevant normal tissues. We generated a reference chromatin accessibility atlas for normal uterine (n=1), ovarian (n=3), and breast (n=1) tissue using DNase-seq data by the Roadmap Epigenomics project⁴ and assembled an atlas of ~397K accessibility regions. We performed motif analysis in each chromatin accessible regions in the common atlas. Then, we examined the patterns of gain or loss of chromatin accessible regions between each pair of tumor types by performing pairwise differential read count analysis on accessible regions. While several FOS family motifs and SMARCC1 are enriched both in normal uterus vs. ovary as well as in the comparison of uterine serous vs. ovarian serous, in most cases the motifs identified by differential accessibility in cancer cell lines did not arise from the tissue of origin based on available normal tissue accessibility data (**Supplementary Figure 4**)

Supplementary Figure 4: Pairwise comparison of transcription factor motifs enriched in differentially accessible regions in normal tissues Uterus vs Ovary. Volcano plot showing effect size versus $-\log_{10}(P)$, using a Bonferroni correction to adjust P values for each plot. TF symbol annotations are written where the adjusted $P < 10^{-5}$. The foreground occurrence is the number of peaks containing a particular TF motif within the group of differential accessible peaks according to \log_2 fold-change read counts, respectively. The background occurrence is the number of peaks containing a particular TF motif found among all the differentially accessible peaks.

I apologize if it is in the manuscript and I didn't see it. Other than this, I think the manuscript is appropriate for publication in Nat Comm

Reviewer #3 (Remarks to the Author):

The manuscript presents a new computational method, PSIONIC, that combines chromatin accessibility with gene expression data to better understand the effect of enhancers in transcriptional programs in cancer. The idea of leveraging epigenomic data, as well as the use of Multi task learning, is novel and the results presented in the article are promising. The authors go one step further and experimentally validate the ability of two PSIONIC predicted transcription factors (ETV6 and MITF) to influence prognostic outcome. This article has the potential of influencing personalized cancer therapy. The authors performed extensive validation studies in response to the first round of reviews. I find that they have adequately addressed the critiques and therefore I recommend this manuscript for publication.

References

- 1 Korch, C. *et al.* DNA profiling analysis of endometrial and ovarian cell lines reveals misidentification, redundancy and contamination. *Gynecol Oncol* **127**, 241-248, doi:10.1016/j.ygyno.2012.06.017 (2012).
- 2 Yamada, K. *et al.* Establishment and characterization of JHUCS-1 cell line derived from carcinosarcoma of the human uterus. *Hum Cell* **17**, 139-144 (2004).
- 3 Yuan, Y. *et al.* Establishment and characterization of cell lines derived from uterine malignant mixed Mullerian tumor. *Gynecol Oncol* **66**, 464-474, doi:10.1006/gyno.1997.4802 (1997).
- 4 Roadmap Epigenomics, C. *et al.* Integrative analysis of 111 reference human epigenomes. *Nature* **518**, 317-330, doi:10.1038/nature14248 (2015).